# EBV miRNAs BART11 and BART17-3p promote immune escape through the enhancer-mediated transcription of *PD-L1*

Jie Wang[1,2], Junshang Ge[2], Yian Wang[2], Fang Xiong[3], Jiayue Guo[2], Xianjie Jiang[2], Lishen Zhang[2], Xiangying Deng[2], Zhaojian Gong[4], Shanshan Zhang[3], Qijia Yan[3], Yi He[1], Xiayu Li[5], Lei Shi[2,4], Can Guo [2], Fuyan Wang[2], Zheng Li[2], Ming Zhou[2], Bo Xiang [2], Yong Li [6], Wei Xiong [1,2✉] & Zhaoyang Zeng [1,2✉]

Epstein-Barr virus (EBV) is reportedly the first identified human tumor virus, and is closely related to the occurrence and development of nasopharyngeal carcinoma (NPC), gastric carcinoma (GC), and several lymphomas. PD-L1 expression is elevated in EBV-positive NPC and GC tissues; however, the specific mechanisms underlying the EBV-dependent promotion of PD-L1 expression to induce immune escape warrant clarification. EBV encodes 44 mature miRNAs. In this study, we find that EBV-miR-BART11 and EBV-miR-BART17-3p upregulate the expression of PD-L1 in EBV-associated NPC and GC. Furthermore, EBV-miR-BART11 targets *FOXP1*, EBV-miR-BART17-3p targets *PBRM1*, and FOXP1 and PBRM1 bind to the enhancer region of PD-L1 to inhibit its expression. Therefore, EBV-miR-BART11 and EBV-miR-BART17-3p inhibit FOXP1 and PBRM1, respectively, and enhance the transcription of *PD-L1* (*CD274*, http://www.ncbi.nlm.nih.gov/gene/29126), resulting in the promotion of tumor immune escape, which provides insights into potential targets for EBV-related tumor immunotherapy.

[1] NHC Key Laboratory of Carcinogenesis and Hunan Key Laboratory of Cancer Metabolism, Hunan Cancer Hospital and the Affiliated Cancer Hospital of Xiangya School of Medicine, Central South University, Changsha, Hunan, China. [2] Key Laboratory of Carcinogenesis and Cancer Invasion of the Chinese Ministry of Education, Cancer Research Institute and School of Basic Medicine Sciences, Central South University, Changsha, Hunan, China. [3] Department of Stomatology, Xiangya Hospital, Central South University, Changsha, Hunan, China. [4] Department of Oral and Maxillofacial Surgery, The Second Xiangya Hospital, Central South University, Changsha, Hunan, China. [5] Hunan Key Laboratory of Nonresolving Inflammation and Cancer, Disease Genome Research Center, The Third Xiangya Hospital, Central South University, Changsha, Hunan, China. [6] Department of Medicine, Dan L Duncan Comprehensive Cancer Center, Baylor College of Medicine, Houston, TX, USA. ✉email: xiongwei@csu.edu.cn; zengzhaoyang@csu.edu.cn

Epstein–Barr virus (EBV) is reportedly the first identified human virus associated with cancer and is associated with the development of nasopharyngeal carcinoma (NPC), gastric carcinoma (GC), and several lymphomas[1,2]. It infects over 90% of the world's adult's population, establishing a persistent latent infection in the host that is sustained by the establishment of a balance between EBV and the host immune system.

Tumor immune escape is important for tumor survival and development. Tumor cells undergo growth and metastasis using mechanisms that help them avoid recognition and attack by the immune system. EBV infection is closely associated with tumor immune escape. Protein products of EBV such as *LMP1*, *EBNA1*, and *EBNA2* can regulate PD-L1 expression to promote immune escape[3–6]. EBV is considered the first identified human virus that encodes microRNAs (miRNAs)[7]. A total of 44 EBV-encoded miRNAs (EBV miRNAs) are involved in cell proliferation, apoptosis, and transformation, and in enabling infected cells to escape immune recognition by targeting host or viral mRNAs. Of them, BART cluster miRNAs are highly expressed in EBV-associated epithelial tumor tissues. However, the role of BART cluster miRNAs in PD-L1 expression has not been clarified.

The immune checkpoint programmed cell death protein 1 (PD-1) and its ligand PD-L1 are crucial for tumor immune escape and immunotherapy. PD-L1 is expressed in a variety of tumors[8,9]. PD-L1 can enable the formation of a barrier on the tumor cell surface, which establishes interaction with the T-cell surface receptor PD-1 to inhibit the cytotoxic effect of the T cell[10]. Accordingly, anti-PD-1/PD-L1 therapy can relieve immunosuppression by facilitating reactivation of immune cells, thereby significantly improving the therapy response of patients with advanced tumors. However, clinical studies have shown that anti-PD-1/PD-L1 therapy efficacy is not ideal for treating solid tumors[11,12].

In this work, we investigate the mechanisms regulating PD-L1 expression and assess the regulatory functions of EBV-encoded miRNAs in PD-L1 in NPC and EBV-associated gastric carcinoma (EBVaGC) by targeting FOXP1 and PBRM1.

## Results

### EBV-miR-BART11 and EBV-miR-BART17-3p upregulate PD-L1 expression in NPC and GC cells.
To identify the role of PD-L1 in the development of EBV-associated epithelial cancers, *PD-L1* expression was analyzed using the genome-wide gene expression profile data available for NPC and GC (GSE12452[13] and GSE65801[14]). The expression of *PD-L1* was higher in NPC and GC tissues than that in normal control tissues (Supplementary Fig. 1a). Using the data from TCGA[2] and GSE51575[15], we analyzed the correlation between *PD-L1* expression and EBV infection and found that *PD-L1* expression in EBV-positive GC tissues was significantly higher than that in EBV-negative GC tissues (Supplementary Fig. 1b).

To confirm the relationship between EBV infection and *PD-L1* expression, we examined *PD-L1* expression in 82 NPC samples (42 EBV-positive and 40 EBV-negative cases) and 31 non-tumor nasopharyngeal epithelium (NPE) clinical tissues by performing quantitative real-time PCR (qRT-PCR). *PD-L1* exhibited significantly high expression in NPC tissues and was positively associated with EBV infection (Fig. 1a). IHC experiments further confirmed the high expression of PD-L1 in 52 additional NPC samples (39 EBV-positive and 13 EBV-negative cases) compared with that in 36 non-tumor NPE tissues (Fig. 1b), and in 40 GC tissues (25 EBV-positive and 15 EBV-negative) compared with that in 20 normal gastric mucosa tissues (Fig. 1c).

Next, we selected the EBV-negative immortalized normal nasopharyngeal epithelial cell line NP69, EBV-positive NPC cell line C666-1, EBV-negative NPC cell line HONE1, normal gastric epithelial cell line GES-1, EBV-positive GC cell line SNU-719, EBV-negative GC cell line AGS, and stably infected EBV (Akata-derived) HONE1 cell line and AGS cells (HONE1-EBV and AGS-EBV) for further study. qRT-PCR analysis of *EBER1* was performed to identify EBV infection. *EBER1* was not expressed in EBV-negative cells (NP69, HONE1, GES-1, and AGS) but was highly expressed in EBV-positive cells (C666-1, HONE1-EBV, SNU-719, and AGS-EBV) (Supplementary Fig. 1c). *PD-L1* showed significantly higher expression in EBV-positive cells than that in EBV-negative cells, as observed via qRT-PCR and western blotting (Supplementary Fig. 1c, d).

EBV derived from B95-8 is widely used as an EBV strain[16]. EBV-negative NPC cell lines HNE2, CNE2, and HONE1 were infected with Akata-derived or B95-8-derived EBV viruses. Akata-derived EBV exposure significantly promoted *PD-L1* expression. However, its expression did not change significantly in B95-8-derived EBV-infected cells, as evidenced via qRT-PCR and western blotting (Supplementary Fig. 1d, e). Compared with Akata-derived EBV, the EBV B95-8 strain lacks the 12-kb genomic region BamH I A rightward transcripts (BART) locus wherein several BART miRNAs are encoded (Supplementary Fig. 1e). This finding suggested that the BART cluster miRNAs in the B95-8 deletion region might be important for EBV in the regulation of *PD-L1* expression.

A series of EBV miRNAs mimics in the B95-8 deletion region were transfected into EBV-negative HONE1 and AGS cells (Supplementary Fig. 1f). Only EBV-miR-BART17-3p, EBV-miR-BART11-3p, and EBV-miR-BART11-5p mimics significantly upregulated *PD-L1* expression in HONE1 and AGS cells (Fig. 1f and Supplementary Fig. 1f–h). Meanwhile, *PD-L1* expression was inhibited in EBV-positive HONE1-EBV, AGS-EBV, C666-1, and SNU-719 cells after the inhibitors of EBV-miR-BART17-3p, EBV-miR-BART11-3p, or EBV-miR-BART11-5p were used (Fig. 1f and Supplementary Fig. 1h). These results suggested that EBV might induce *PD-L1* expression through EBV-miR-BART17-3p, EBV-miR-BART11-3p, and EBV-miR-BART11-5p.

IFN-γ plays an important role in PD-L1 expression upregulation in the tumor microenvironment. Therefore, we explored whether IFN-γ could cooperate with EBV to promote *PD-L1* expression after IFN-γ (10 ng/mL) stimulation and found that IFN-γ demonstrated functions in both EBV-negative and EBV-positive cells; however, EBV infection further enhanced the induction of *PD-L1* expression by IFN-γ (Supplementary Fig. 2a). Furthermore, treatment with EBV-miR-BART11 or EBV-miR-BART17-3p mimics accelerated *PD-L1* expression after IFN-γ stimulation, while treatment with their inhibitors reduced the expression (Supplementary Fig. 2b).

### EBV-miR-BART17-3p upregulates PD-L1 expression by targeting *PBRM1*.
Next, we explored the possible molecular mechanisms by which EBV-miR-BART17-3p, EBV-miR-BART11-3p, and EBV-miR-BART11-5p regulate PD-L1. Using miRNA databases such as miRanda and RNAhybrid, 26 potential targets of EBV-miR-BART17-3p with low free energy were selected (Supplementary Fig. 3a). EBV-miR-BART17-3p is highly expressed in NPC (Supplementary Fig. 3b) in the NPC miRNA datasets (GSE32960 and GSE36682); hence, we selected target genes of EBV-miR-BART17-3p whose expression might be downregulated in NPC tissues. The gene expression profile data for NPC (GSE12452 and GSE64634) were analyzed and the expression of a total of 511 genes was found to be significantly downregulated in NPC tissues (Supplementary Fig. 3c, d). Of these genes, *PBRM1* and *STK40* were predicted as potential targets of EBV-miR-BART17-3p (Supplementary Fig. 3e), and their

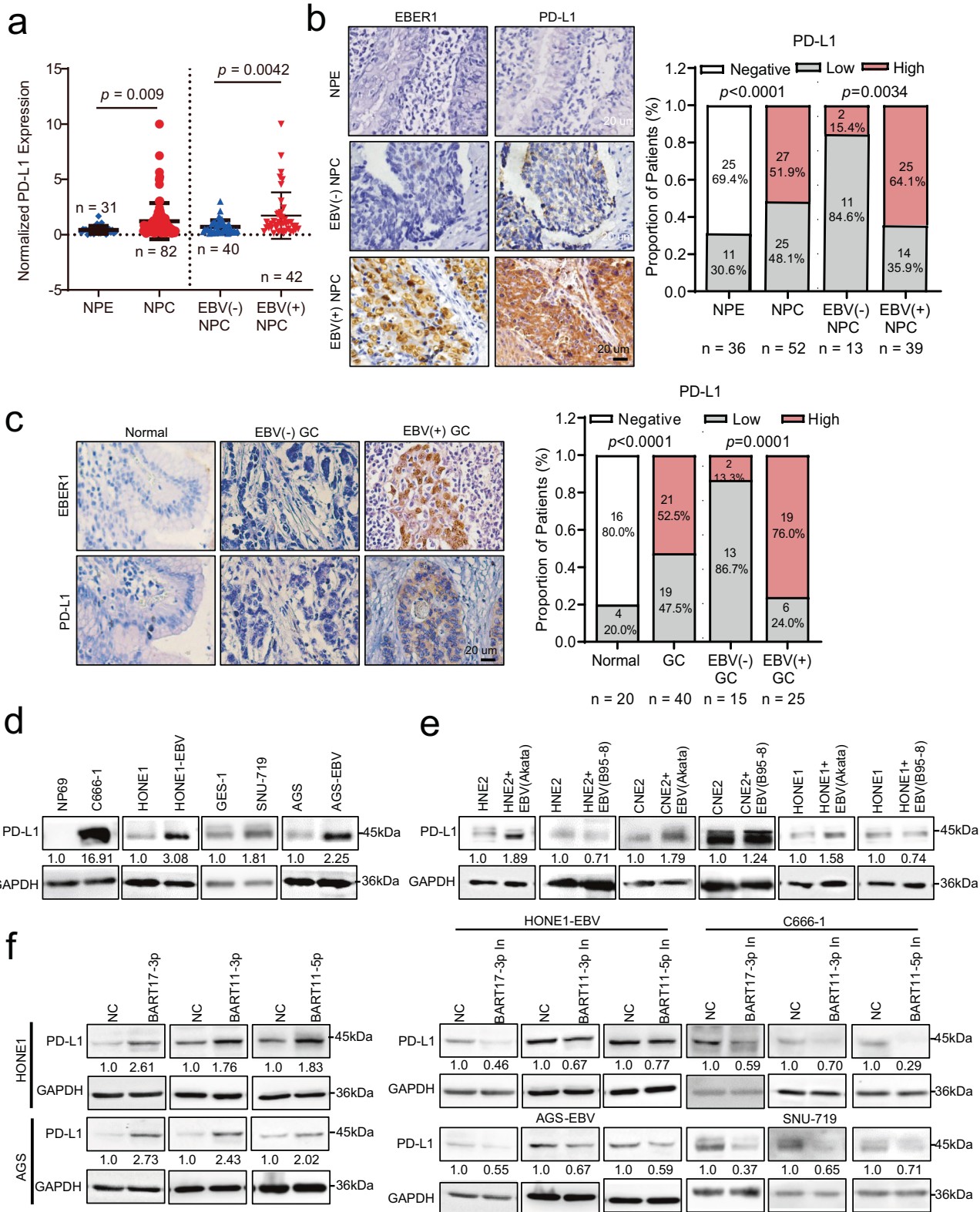

expression was also significantly and negatively correlated with that of *PD-L1* in NPC tissues (Supplementary Fig. 3f).

To identify whether *PBRM1* and *STK40* were indeed EBV-miR-BART17-3p targets, EBV-miR-BART17-3p expression was confirmed in HONE1-EBV and AGS-EBV cells using RNA FISH. EBV-miR-BART17-3p was highly expressed in HONE1-EBV and AGS-EBV cells but was not expressed in HONE1 and AGS

(Supplementary Fig. 4a). Transfection of EBV-miR-BART17-3p mimics into HONE1 and AGS cells showed that PBRM1 expression was inhibited, whereas EBV-miR-BART17-3p inhibitors promoted PBRM1 expression in HONE1-EBV, AGS-EBV, C666-1, and SNU-719 cells, as identified via qRT-PCR and western blotting (Supplementary Fig. 4b, c). However, *STK40* expression remained unchanged in the presence of EBV-miR-BART17-3p (Supplementary

**Fig. 1 EBV-miR-BART11 and EBV-miR-BART17-3p upregulate PD-L1 expression in NPC and GC. a** The expression of *PD-L1* mRNA was evaluated in 82 NPC samples (40 EBV-negative and 42 EBV-positive) and 31 NPE tissue samples via qRT-PCR. NPC, nasopharyngeal carcinoma; NPE, nasopharyngeal epithelial. **b** The expression of PD-L1 and EBER1 was examined in 52 NPC samples (13 EBV-negative and 39 EBV-positive) and 36 NPE tissue samples via IHC or ISH. Magnification: ×400; scale bars = 20 μm. The figure on the right shows the results for the statistical analysis of PD-L1 expression in NPC samples and NPE tissues ($p < 0.0001$) and the correlation between PD-L1 expression and EBV in NPC samples ($p = 0.0034$). **c** The expression of PD-L1 and EBER1 was examined in 40 GC samples (15 EBV-negative and 25 EBV-positive) and 20 normal gastric mucosa tissues via IHC or ISH. Magnification: ×400; scale bars = 20 μm. The figure on the right shows the results for the statistical analysis of PD-L1 expression in GC samples and normal gastric mucosa tissues ($p < 0.0001$) and the correlation between PD-L1 expression and EBV in GCs ($p = 0.0001$). **d** Western blotting was performed to quantify the protein level of PD-L1 in EBV-negative immortalized NPE cell line NP69 and EBV-positive NPC cell line C666-1, NPC cells HONE1 and HONE1-EBV stably transfected with EBV (Akata-derived), normal gastric epithelial cell line GES-1, GC cell lines including SNU-719, AGS, and AGS-EBV stably transfected with EBV (Akata-derived). GAPDH was used as an internal control. **e** Western blotting was used to detect the protein level of PD-L1 in HNE2, CNE2, and HONE1 cells that were infected with EBV virions derived from Akata or B95-8. GAPDH was used as an internal control. **f** Western blotting was used to detect the expression of PD-L1 in HONE1 and AGS cells transfected with EBV-miR-BART17-3p, EBV-miR-BART11-3p, or EBV-miR-BART11-5p mimics, or EBV-positive HONE1-EBV, AGS-EBV, C666-1, and SNU-719 cells transfected with EBV-miR-BART17-3p, EBV-miR-BART11-3p, or EBV-miR-BART11-5p inhibitors. GAPDH was used as an internal control. **b**, **c** are calculated by *F*-test. Source data are provided as a Source Data file.

Fig. 4d). The luciferase activity of the *PBRM1* wild-type vector was reduced; however, no effect on the activity of the *PBRM1* mutant vector in HONE1 and AGS cells (Fig. 2a) was observed after co-transfection of luciferase reporter vectors containing wild-type or mutant sequences for the 3′-UTR of *PBRM1* (7562–7591 bp) (Supplementary Fig. 4e) and EBV-miR-BART17-3p mimics. Inhibition of EBV-miR-BART17-3p in HONE1-EBV, AGS-EBV, C666-1, and SNU-719 cells enhanced the luciferase activity of the *PBRM1* wild-type vector but did not affect the *PBRM1* mutant vector (Fig. 2a). These results indicated that EBV-miR-BART17-3p could directly bind to the *PBRM1* 3′-UTR, resulting in the downregulation of *PBRM1* expression. To further confirm whether EBV-miR-BART17-3p could directly bind to the *PBRM1* 3′-UTR, biotin-labeled or unlabeled BART17-3p probes were transfected into the HONE1 and AGS cell lines, respectively. RNA pull-down and qRT-PCR assays showed that biotin-labeled BART17-3p probes enriched the 3′-UTR sequence of *PBRM1* (Fig. 2b and Supplementary Fig. 4f). AGO2 can mediate miRNA binding to the 3′-UTR of mRNA. Therefore, RIP using the anti-AGO2 antibody was performed in HONE1 and AGS cells after transfection with EBV-miR-BART17-3p mimics, and the 3′-UTR sequence of *PBRM1* after pull-down of AGO2 was increased significantly (Fig. 2c and Supplementary Fig. 4g). Thus, EBV-miR-BART17-3p directly bound to the 3′-UTR sequence of *PBRM1* and mediated the degradation of *PBRM1* mRNA through the RNA-induced silencing (RISC) complex.

PBRM1 is a transcriptional regulatory protein that can inhibit the transcription of downstream genes[17,18]. In NPC and GC cells, PBRM1 overexpression reduced PD-L1 expression and *PBRM1* knockdown induced PD-L1 expression (Fig. 2d). EBV-miR-BART17-3p overexpression in HONE1 and AGS cells inhibited PBRM1 expression and upregulated PD-L1 expression, and PBRM1 attenuated PD-L1 expression induced by EBV-miR-BART17-3p in EBV-negative cells after co-transfection with EBV-miR-BART17-3p mimics and the PBRM1 overexpression vector (Fig. 2d). Furthermore, EBV-miR-BART17-3p inhibitors induced PBRM1 expression and reduced PD-L1 expression in HONE1-EBV, AGS-EBV, C666-1, and SNU-719 cells, and this function of EBV-miR-BART17-3p inhibitors was reversed after *PBRM1* knockdown (Fig. 2d). Similar results were also obtained with qRT-PCR assays (Supplementary Fig. 5a), immunofluorescence staining of HONE1 cells (Fig. 2e), and flow cytometry analysis of HONE1 and HONE1-EBV cells (Fig. 2f and Supplementary Fig. 5b). These data indicated that EBV-miR-BART17-3p promoted *PD-L1* expression by targeting and inhibiting *PBRM1* expression in EBV-associated cancers.

**EBV-miR-BART11 upregulates PD-L1 expression by targeting *FOXP1*.** EBV-miR-BART11-3p and EBV-miR-BART11-5p were

highly expressed in the NPC miRNA datasets (GSE32960 and GSE36682) (Supplementary Fig. 6a), and might indirectly upregulate *PD-L1* expression through their target genes. Both EBV-miR-BART11-3p and EBV-miR-BART11-5p were processed by the pre-EBV-miR-BART11 RNA and could bind to the 3′-UTR of *FOXP1* and inhibit its expression in NPC[19]. Therefore, we confirmed whether EBV-miR-BART11-3p and EBV-miR-BART11-5p regulated *PD-L1* expression by targeting *FOXP1*. *FOXP1* expression was found to be significantly downregulated and negatively correlated with *PD-L1* expression as per the GSE12452 database (Supplementary Fig. 6b). Both EBV-miR-BART11-3p and EBV-miR-BART11-5p were highly expressed in HONE1-EBV and AGS-EBV cells and undetectable in HONE1 and AGS cells by RNA FISH (Supplementary Fig. 6c). Transfection of EBV-miR-BART11 mimics inhibited *FOXP1* expression in HONE1 and AGS cells, and that of EBV-miR-BART11 inhibitors promoted *FOXP1* expression in HONE1-EBV, AGS-EBV, C666-1, and SNU-719 cells, as determined via qRT-PCR and western blotting (Supplementary Fig. 6d, e).

EBV-miR-BART11-3p and EBV-miR-BART11-5p overexpression also inhibited the reporter activity of the *FOXP1* 3′-UTR in HONE1 and AGS cells, and inhibition of EBV-miR-BART11-3p and EBV-miR-BART11-5p enhanced the luciferase activity in HONE1-EBV, AGS-EBV, C666-1, and SNU-719 cells (Fig. S6f). The biotin-labeled EBV-miR-BART11-3p and EBV-miR-BART11-5p probes could enrich the 3′-UTR sequence of *FOXP1* using RNA pull-down assays followed by qRT-PCR (Fig. 3a and Supplementary Fig. 7a). The 3′-UTR sequence of *FOXP1* was also immunoprecipitated by using the anti-AGO2 antibody in HONE1 and AGS cells after transfection of EBV-miR-BART11-3p or EBV-miR-BART11-5p mimics via RIP experiments using the anti-AGO2 antibody (Fig. 3b and Supplementary Fig. 7b). Thus, EBV-miR-BART11 (BART11-3p and BART11-5p) could bind directly to the 3′-UTR sequence of *FOXP1* and mediated the degradation of the *FOXP1* mRNA via the RISC complex.

FOXP1 is also a transcriptional repressor[20,21]. Overexpression of FOXP1 leads to a decrease in *PD-L1* expression in NPC and GC cells, as evidenced via qRT-PCR (Supplementary Fig. 7c), western blotting (Fig. 3c), immunofluorescence staining of HONE1 cells (Fig. 3d), and flow cytometry assays of HONE1 and HONE1-EBV cells (Fig. 3e and Supplementary Fig. 7d), whereas *FOXP1* knockdown upregulated PD-L1 expression. EBV-miR-BART11 overexpression inhibited FOXP1 and promoted *PD-L1* expression. FOXP1 reduced PD-L1 expression induced via EBV-miR-BART11 mimics in EBV-negative cells after co-transfection with EBV-miR-BART11 mimics and the FOXP1 overexpression vector. Both EBV-miR-BART11 inhibitors induced *FOXP1* expression and reduced *PD-L1* expression in EBV-positive cells, and *PD-L1* expression was restored after

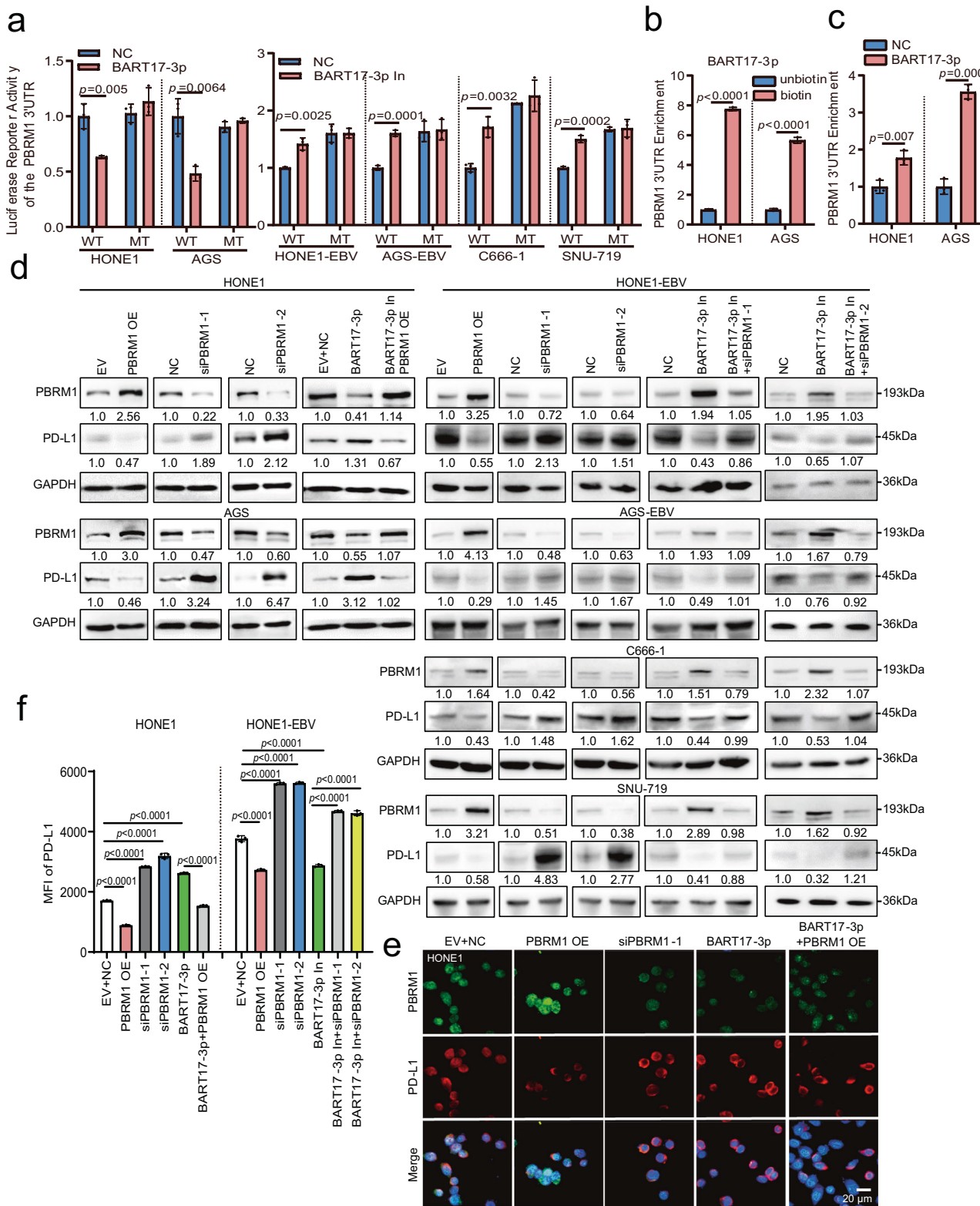

*FOXP1* knockdown. Taken together, the data indicated that EBV-miR-BART11 promoted *PD-L1* expression by targeting and inhibiting *FOXP1* expression in EBV-associated cancers.

**FOXP1 and PBRM1 inhibit *PD-L1* transcription by binding to its enhancer region.** To identify whether the interaction and cooperation between FOXP1 and PBRM1 could inhibit PD-L1 expression, co-immunoprecipitation (Co-IP) experiments were

performed and showed that FOXP1 established interactions with PBRM1 in HONE1 and AGS cells (Fig. 4a). Immunofluorescence staining revealed the endogenous co-localization of FOXP1 and PBRM1 (Fig. 4b) based on the use of anti-FOXP1 and PBRM1 antibodies in HONE1 cells. Western blotting results showed that the concomitant overexpression or inhibition of EBV-miR-BART11 and EBV-miR-BART17-3p exerted a more remarkable regulatory effect on PD-L1 than the overexpression or inhibition

**Fig. 2 EBV-miR-BART17-3p upregulates PD-L1 expression by targeting PBRM1. a** HONE1, AGS, HONE-EBV, AGS-EBV, C666-1, and SNU-719 cells were co-transfected with the *PBRM1*-WT or the *PBRM1*-MT vectors and the EBV-miR-BART17-3p mimics or inhibitors. The effect of EBV-miR-BART17-3p on the luciferase reporter activity of the *PBRM1* 3'-UTR is shown. $n = 3$ biologically independent samples. **b** RNA pull-down assays followed by qRT-PCR were performed to examine the binding effect of EBV-miR-BART17-3p on the 3'-UTR of *PBRM1* in HONE1 and AGS cells transfected with the biotin-labeled or unbiotin-labeled EBV-miR-BART17-3p probes. $n = 3$ biologically independent samples. **c** After transfection of EBV-miR-BART17-3p mimics or negative control into EBV-negative HONE1 and AGS cells, anti-AGO2 antibody was used for the RIP experiment followed by qRT-PCR to detect the binding of EBV-miR-BART17-3p to the *PBRM1* 3'-UTR via AGO2. **d** Western blotting was used to detect whether EBV-miR-BART17-3p could regulate the PD-L1 protein via PBRM1 in the HONE1, AGS, HONE1-EBV, AGS-EBV, C666-1, and SNU-719 cells transfected with the PBRM1 overexpression vector, siPBRM1, EBV-miR-BART17-3p mimics or inhibitors, or co-transfected with EBV-miR-BART17-3p mimics and the PBRM1 overexpression vector, or EBV-miR-BART17-3p inhibitors and siPBRM1. GAPDH was used as an internal control. **e** Immunofluorescence assays were used to identify PD-L1 expression in HONE1 cells transfected with the PBRM1 overexpression vector, siPBRM1, EBV-miR-BART17-3p mimics, or co-transfected EBV-miR-BART17-3p mimics and the PBRM1 overexpression vector. PBRM1: green; PD-L1: red; merge: signal superimposed image of DAPI, PBRM1, and PD-L1; magnification: ×400; scale bars = 20 μm. **f** Flow cytometry analysis of PD-L1 expression in HONE1 and HONE1-EBV cells transfected with the PBRM1 overexpression vector, siPBRM1, EBV-miR-BART17-3p mimics or inhibitors, or co-transfected with EBV-miR-BART17-3p mimics and the PBRM1 overexpression vector or EBV-miR-BART17-3p inhibitors and siPBRM1. Each group was analyzed using 3 independent replicates, and the mean fluorescence intensity (MFI) of PD-L1 was calculated. The original results are shown in Supplementary Fig. 5b. Data are presented as mean ± s.d., and *p*-values are calculated by unpaired two-sided *t*-test in **a**, **b**, **f**. Source data are provided as a Source Data file.

of EBV-miR-BART11 or EBV-miR-BART17-3p alone. The simultaneous overexpression or inhibition of FOXP1 and PBRM1 enhanced PD-L1 expression more significantly than FOXP1 or PBRM1 overexpression or inhibition alone. However, over-expression or inhibition of EBV-miR-BART11 demonstrated no effect on PBRM1 expression, and EBV-miR-BART17-3p did not regulate FOXP1 expression. FOXP1 and PBRM1 did not affect each other's expression (Fig. 4c).

The expression levels of EBV-miR-BART11-3p, EBV-miR-BART11-5p, EBV-miR-BART17-3p, FOXP1, PBRM1, and PD-L1 were determined using samples of NPC and EBV-associated gastric carcinoma tissues by performing IHC or ISH methods. FOXP1 and PBRM1 expression levels were low, while EBV-miR-BART11-3p, EBV-miR-BART11-5p, and EBV-miR-BART17-3p were highly expressed in NPC and gastric carcinoma tissues. EBV-miR-BART17-3p was negatively correlated with PBRM1 expression and was positively correlated with PD-L1 expression. EBV-miR-BART11-3p and EBV-miR-BART11-5p expression levels were negatively correlated with FOXP1 expression but were positively correlated with PD-L1 expression (Fig. 4d and Supplementary Fig. 8a–d). We also identified the proportion of $CD3^+$ T cells in the peripheral blood of patients with clinical NPC and found it to be lower than that in normal individuals (Supplementary Fig. 8e). Moreover, in patients with EBV-positive NPC, the proportion of $CD3^+$ T cells was lower in the peripheral blood than that in patients who were EBV-negative (Supplementary Fig. 8e), indicating that EBV-positive patients were more markedly immunosuppressed than EBV-negative patients.

To identify the mechanisms underlying the regulation of PD-L1 expression by FOXP1 and PBRM1, a series of truncated mutants for the *PD-L1* promoter region ($-1940$ to $+87$ bp) were cloned into the PGL3-Basic vector. Results from the luciferase activity assay showed that the region from $-1940$ to $-1567$ bp was critical to the *PD-L1* promoter to be subjected to regulation by PBRM1 and FOXP1 (Supplementary Fig. 9a). The classical promoter region of *PD-L1* is located at a region spanning $-798$ to $+153$ bp[22]; therefore, we hypothesized that PBRM1 and FOXP1 might regulate *PD-L1* expression by enabling binding of the *PD-L1* enhancer but not via the classic *PD-L1* promoter. We examined the H3K27ac and H3K4me1 modifications in the *PD-L1* transcriptional regulatory region (spanning $-20,503$ to $+49,497$ bp) in the ENCODE database and displayed it in UCSC Browser, as these modifications aid opening of the chromatin in the enhancer region[23–25]. Next, we downloaded and reanalyzed the H3K27ac and H3K4me1 modification data from the published ChIP-Seq datasets (GSE95749 for EBV-positive

C666−1 and EBV-negative HNE1)[26]. The modifications H3K27ac and H3K4me1 in EBV-positive C666-1 and EBV-negative HNE1 were compared. The data showed that activated enhancers tend to be enriched for H3K27ac and H3K4me1. The H3K27ac and H3K4me1 peaks in the B and E regions in EBV-positive C666-1 were higher than the peaks in EBV-negative HNE1 (Fig. S9b), indicating that B and E are activated enhancers in C666-1. To identify whether PBRM1 and FOXP1 regulated *PD-L1* expression through binding of the *PD-L1* enhancer, a chromatin conformation capture (3C) experiment was performed. Remote interactions between the B region (spanning $-1940$ to $-1567$ bp) and E region (spanning 43,084 to 44,733 bp) and the *PD-L1* promoter were analyzed (Supplementary Fig. 10a). FOXP1 or PBRM1 inhibition enhanced the remote interactions and FOXP1 or PBRM1 overexpression abrogated these interactions in HONE1 cells (Fig. 5a).

Next, the binding site of FOXP1 and PBRM1 in the *PD-L1* enhancer region was predicted using the JASPAR database and the potential binding sites for FOXP1 were at the B (spanning $-1709$ to $-1695$ bp) and E (spanning $+43,285$ to $+43,296$ bp) sites of *PD-L1* (Supplementary Fig. 10b). The binding site of PBRM1 was not predicted in the *PD-L1* enhancer, indicating that PBRM1 might indirectly bind to the enhancer. The enhancer B and E regions and their corresponding mutant vectors were cloned into luciferase reporter vectors. The luciferase activity assay showed that FOXP1 or PBRM1 inhibition enhanced the activity of the enhancers B, E, and B + E, whereas their overexpression decreased this activity (Fig. 5b). Mutation of the FOXP1 binding site in the fragments B or E caused a significant reduction in fluorescence activity, and overexpression of FOXP1 or PBRM1 did not affect the luciferase activity of the mutations (Fig. 5b). The results suggested that FOXP1 and PBRM1 could regulate *PD-L1* expression through binding to its B and E regions. ChIP experiments further confirmed FOXP1 exerted a strong binding effect on the B and E regions and no effect on the A, C, and D regions was observed; additionally, knockdown of *FOXP1* caused weak binding of FOXP1 to the B and E regions, and knockdown of *PBRM1* exerted no effect on FOXP1 binding to the B and E regions. Moreover, FOXP1 or PBRM1 inhibition decreased the binding of PBRM1 to the B and E regions (Fig. 5c and Supplementary Fig. 10c, d). Taken together, it could be inferred that FOXP1 and PBRM1 could exert effects on the B and E regions of the *PD-L1* enhancer, and PBRM1 could bind to the B and E regions via FOXP1. Additionally, *FOXP1* or *PBRM1* knockdown increased the H3K27ac and H3K4me1 modification levels on the B and E regions. Finally, we confirmed the direct

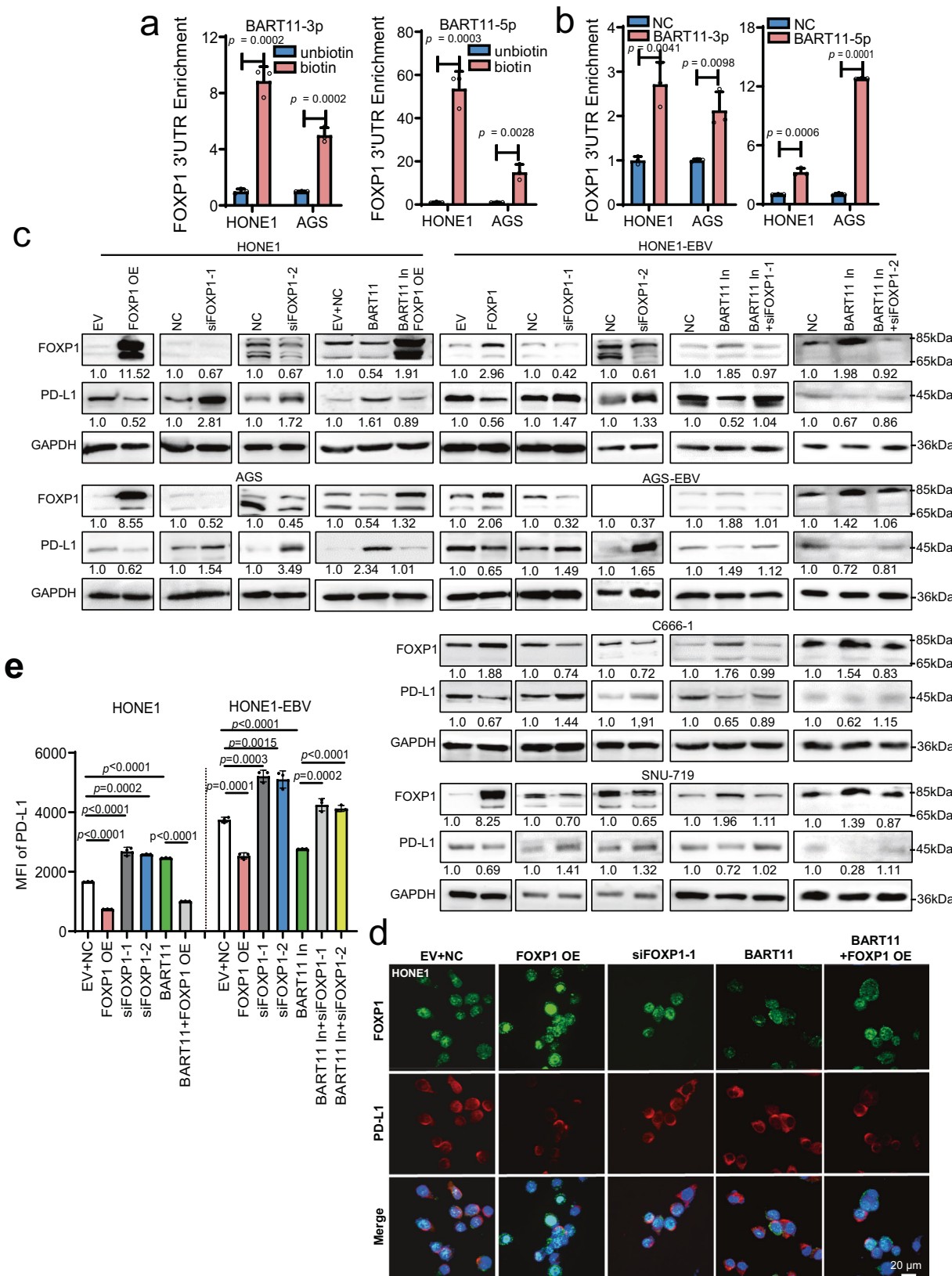

binding of FOXP1 to the B and E sequences of the *PD-L1* enhancer via EMSA experiments, whereas PBRM1 did not show direct binding (Fig. 5d and Supplementary Fig. 10e). FOXP1 and PBRM1 form a complex that recognizes and binds to the B and E regions of PD-L1 through FOXP1, and this binding affects the nearby histone modifications and inhibits PD-L1 expression.

To examine whether EBV-miR-BART11 or EBV-miR-BART17-3p regulated PD-L1 expression through the enhancer region, 3C experiments were performed and it was observed that EBV-miR-BART11 and EBV-miR-BART17-3p promoted the long-range interaction between the B and E regions and the *PD-L1* promoter region in HONE1 cells upon their

**Fig. 3 EBV-miR-BART11 upregulates PD-L1 expression by targeting FOXP1. a** RNA pull-down assays followed by qRT-PCR were performed to examine the binding effect of EBV-miR-BART11-3p or EBV-miR-BART11-5p on the 3′-UTR of *FOXP1* in HONE1 and AGS cells transfected with the biotin-labeled or unbiotin-labeled EBV-miR-BART11-3p or EBV-miR-BART11-5p probes. *n* = 3 biologically independent samples. **b** After transfection of EBV-miR-BART11-3p, EBV-miR-BART11-5p mimics, or negative control into EBV-negative HONE1 and AGS cells, anti-AGO2 antibody was used for RIP analysis followed by qRT-PCR to identify whether EBV-miR-BART11-3p or EBV-miR-BART11-5p could bind the *FOXP1* 3′-UTR via AGO2. *n* = 3 biologically independent samples. **c** Western blotting was used to examine whether EBV-miR-BART11 could regulate the PD-L1 protein via FOXP1 in HONE1, AGS, HONE1-EBV, AGS-EBV, C666-1, and SNU-719 cells transfected with the FOXP1 overexpression vector, siFOXP1, EBV-miR-BART11 mimics or inhibitors, or co-transfected with EBV-miR-BART11 mimics and the FOXP1 overexpression vector, or EBV-miR-BART11 inhibitors and siFOXP1. GAPDH was used as an internal control. **d** Immunofluorescence analysis was performed to identify whether EBV-miR-BART11 could regulate PD-L1 via FOXP1 in HONE1 cells transfected with the FOXP1 overexpression vector, siFOXP1, EBV-miR-BART11 mimics, or co-transfected with EBV-miR-BART11 mimics and the FOXP1 overexpression vector. FOXP1: green; PD-L1: red; merge: signal superimposed image of DAPI, FOXP1, and PD-L1; magnification: ×400; scale bars = 20 μm. **e** Flow cytometry was used to detect PD-L1 expression in HONE1 and HONE1-EBV cells transfected with the FOXP1 overexpression vector, siFOXP1, EBV-miR-BART11 mimics or inhibitors, or co-transfected with EBV-miR-BART11 mimics and the FOXP1 overexpression vector, or EBV-miR-BART11 inhibitors and siFOXP1. Each group was analyzed using three independent replicates, and the mean fluorescence intensity (MFI) of PD-L1 was calculated. The original result is shown in Supplementary Fig. 7d. Data are presented as mean ± s.d., and *p*-values are calculated by unpaired two-sided *t*-test in **a**, **b**, **e**. Source data are provided as a Source Data file.

overexpression. EBV-miR-BART11 and EBV-miR-BART17-3p inhibition reduced their interaction in C666-1 and SNU-719 cells (Supplementary Fig. 10f). Results of both dual-luciferase reporter gene assays and ChIP experiments confirmed that EBV-miR-BART11 or EBV-miR-BART17-3p impacted the binding of FOXP1 and PBRM1 to the *PD-L1* enhancer (Fig. 5e and Supplementary Fig. 10g) and the nearby histone modification (Fig. 5f and Supplementary Fig. 10h).

**FOXP1 establishes interactions with the PBRM1 containing the PBAF complex and inhibits PD-L1 expression**. PBRM1 is one of the main components of the SWI/SNF PBAF complex and participates in the modulation of gene expression by regulating chromatin remodeling[27]. The SWI/SNF chromatin remodeling complex comprises two categories, namely BAF and PBAF. Both share the SMARCA4, SMARCE1, SMARCC1, and β-actin proteins, whereas DPF2 is specific to BAF and PBRM1 is a specific subunit of PBAF[28]. To identify whether FOXP1 established interaction with this complex, Co-IP combined with mass spectrometry was performed and results showed that FOXP1 established interaction with the main components shared by the SWI/SNF complexes such as SMARCE1, SMARCC1, and β-actin (Supplementary Fig. 11a and Supplementary Data 1). Western blotting experiments revealed that FOXP1 could bind to the PBRM1, SMARCA4, SMARCE1, and β-actin components of PBAF but could not bind to the BAF-specific component DPF2 (Fig. 6a). We further investigated the interactions with PBRM1, FOXP1, SMARCA4, SMARCE1, and β-actin, and found that PBRM1 did not bind to DPF2 (Fig. 6b). Correlation analysis of GSE12452 showed that *DPF2* was positively correlated with *PD-L1* expression and was negatively correlated with *PBRM1* expression (Supplementary Fig. 11b). *DPF2* knockdown did not affect FOXP1 and PBRM1 expression but decreased that of PD-L1 (Fig. 6c and Supplementary Fig. 11c–e).

*FOXP1* or *PBRM1* knockdown in NPC and GC cells induced the binding of DPF2 to the *PD-L1* enhancer regions B and E (Fig. 6d). *DPF2* knockdown reduced its binding to the PD-L1 enhancer regions B and E and the H3K27ac and H3K4me1 modifications and promoted the binding of FOXP1 and PBRM1 to the *PD-L1* enhancer regions B and E (Fig. 6e and Supplementary Fig. 11f). Results from the luciferase reporter gene experiments showed that *DPF2* knockdown weakened the luciferase activity of the *PD-L1* enhancer regions B and E (Fig. 6f). These results indicated that the BAF-specific subunit DPF2 might regulate PD-L1 expression, and PBRM1 and DPF2 might compete for binding to the *PD-L1* enhancers B and E by assembling the SWI/SNF complexes PBAF or BAF. FOXP1

establishes interactions with PBRM1 to promote the binding of PBAF to the *PD-L1* enhancer and inhibits PD-L1 expression.

**EBV-miR-BART11 and EBV-miR-BART17-3p induce T cell apoptosis by promoting PD-L1 expression in NPC and gastric carcinoma cells**. PD-L1 can induce T-cell apoptosis or inhibit T-cell activity through PD-1 present on the surface of T cells to promote immune escape of tumor cells. To identify whether EBV-miR-BART11 and EBV-miR-BART17-3p induced T-cell apoptosis by promoting PD-L1 expression, a high-content screening system was used to track the activity status of T cells in real-time. First, HONE1 and AGS cells transfected with EBV-miR-BART11 and EBV-miR-BART17-3p mimics were subjected to staining procedures with the CM-DiI live-cell fluorescent dye (red) and were subsequently co-cultured with activated human primary T cells labeled with the live-cell fluorescent dye CMFDA (green) at a ratio of 1:10. The fluorescence intensity of primary T cells in the EBV-miR-BART11 and EBV-miR-BART17-3p group decreased significantly at a faster rate than that of the control cells (Fig. 7a and Supplementary Movies 1–4). The addition of sufficient PD-L1 blocking antibody to the cell culture medium decreased the ability of EBV-miR-BART11 and EBV-miR-BART17-3p to promote T cell apoptosis (Supplementary Fig. 12a). Hoechst dye (blue) staining performed using tumor cells showed that EBV-miR-BART11 and EBV-miR-BART17-3p overexpression decreased tumor cell apoptosis, indicating that EBV-miR-BART11 and EBV-miR-BART17-3p inhibited the cytotoxic effect of T cells exerted on tumor cells (Supplementary Fig. 12b). Next, the apoptosis of primary T cells in a co-culture with tumor cells was detected via flow cytometry and inhibition of EBV-miR-BART11 and EBV-miR-BART17-3p in HONE1-EBV and AGS-EBV cells reduced the degree of T-cell apoptosis; additionally, the miRNA mimics significantly influenced the degree of T-cell apoptosis in HONE1 and AGS cells (Fig. 7b and Supplementary Fig. 13). Use of a sufficient proportion of PD-L1 blocking antibody reduced the effects of EBV-miR-BART11 and EBV-miR-BART17-3p in HONE1 and AGS cells on T-cell apoptosis (Fig. 7b and Supplementary Fig. 14). When EBV-miR-BART11 and EBV-miR-BART17-3p were overexpressed in T cells, EBV-miR-BART11 and EBV-miR-BART17-3p showed no effect on the degree of T-cell apoptosis (Supplementary Fig. 15), which further confirmed that EBV-miR-BART11 and EBV-miR-BART17-3p could not induce T cell apoptosis directly, but could induce effects by upregulating PD-L1 expression.

T cells can secrete IFN-γ and other cytokines to eliminate cancerous cells. We examined IFN-γ levels in T cells in the co-cultured state and found that EBV-miR-BART11 and EBV-miR-

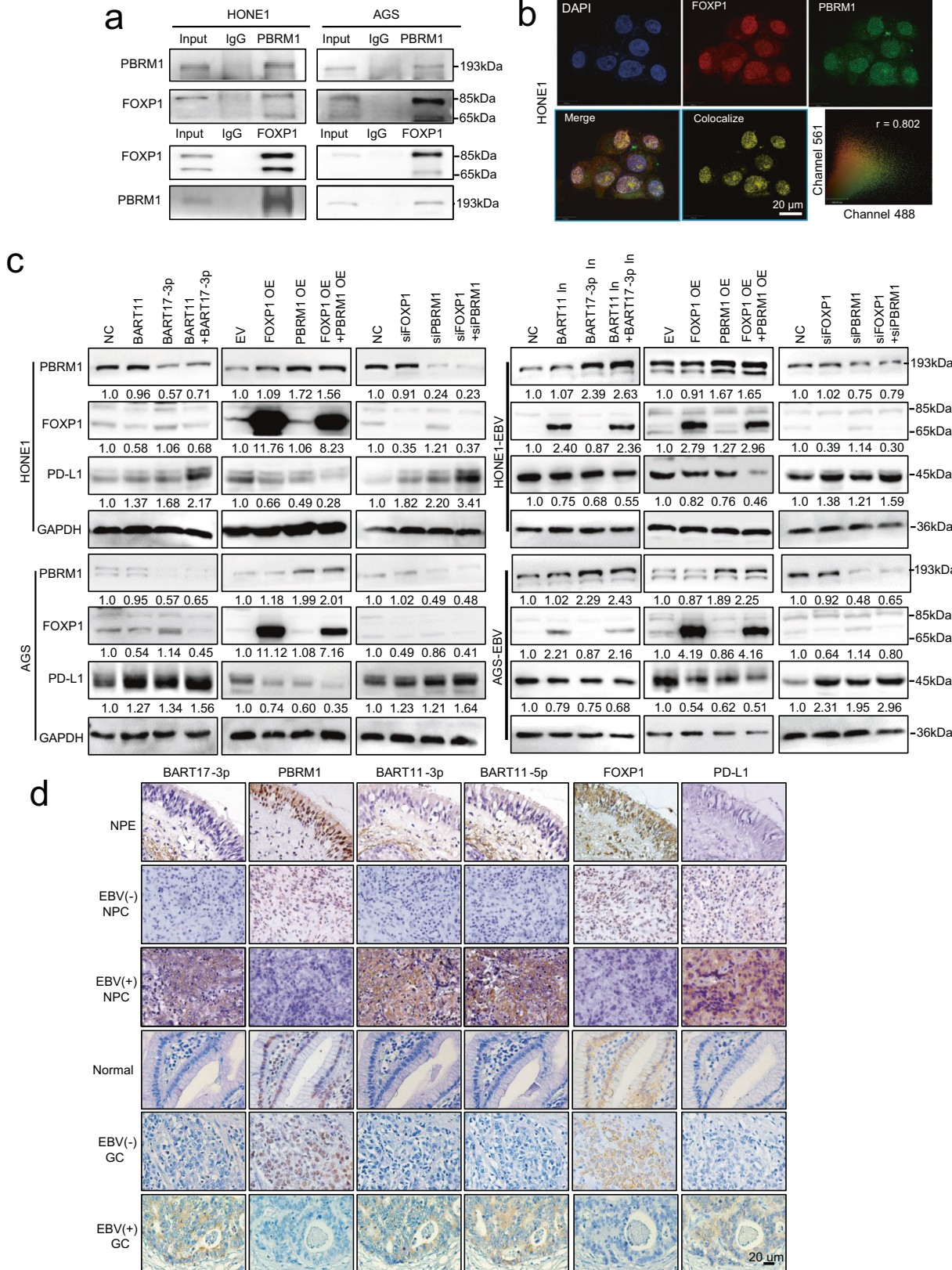

BART17-3p overexpression decreased IFN-γ levels of T cells co-cultured with HONE1 and AGS cells. Conversely, EBV-miR-BART11 and EBV-miR-BART17-3p inhibition promoted the IFN-γ level in HONE1-EBV and AGS-EBV cells (Fig. 7c–e and Supplementary Fig. 16). Therefore, EBV-miR-BART11 and EBV-miR-BART17-3p can effectively reduce the production of IFN-γ

by T cells that surround tumor cells, resulting in the inhibition of their cytotoxic effects.

IFN-γ can inhibit T cells by inducing *PD-L1* expression in the cells surrounding the tumor cells. To ascertain whether EBV-miR-BART11 and EBV-miR-BART17-3p affected IFN-γ and played roles in the regulation of T-cell apoptosis, flow cytometry

**Fig. 4 FOXP1 inhibits PD-L1 expression via interaction with PBRM1. a** Co-immunoprecipitation assays were performed to detect the interaction established between FOXP1 and PBRM1 in HONE1 and AGS cells using anti-FOXP1 and anti-PBRM1 antibodies. **b** Immunofluorescence assays were performed to detect the endogenous co-localization of FOXP1 and PBRM1 in HONE1 cells using anti-FOXP1 and anti-PBRM1 antibodies. DAPI-stained nuclei: blue; FOXP1: red; PBRM1: green; merge: superimposed signals of DAPI, FOXP1, and PBRM1; colocalize: yellow. Scatter analysis shows the signals for channels 561 (FOXP1) and 488 (PBRM1). The Pearson's correlation coefficient of colocalization is 0.802. Magnification: ×400; scale bars = 20 μm. **c** Western blotting assays were performed to detect the regulation of FOXP1 and PBRM1 and the effects on the expression of PD-L1 in HONE1, AGS, HONE1-EBV, and AGS-EBV cells transfected with EBV-miR-BART11 or EBV-miR-BART17-3p mimics or inhibitors, or co-transfected with EBV-miR-BART11 and EBV-miR-BART17-3p mimics or inhibitors, or the FOXP1 or PBRM1 overexpression vector or siRNAs. GAPDH was used as an internal control. **d** The expression of PBRM1, FOXP1, and PD-L1 using IHC, and the expression of EBV-miR-BART17-3p, EBV-miR-BART11-3p, and EBV-miR-BART11-5p using ISH were analyzed in 52 NPCs compared with 36 NPEs, and in 40 GCs compared with 20 normal gastric mucosa tissues. Magnification: ×400; scale bars = 20 μm. Source data are provided as a Source Data file.

was performed and EBV-miR-BART11 and EBV-miR-BART17-3p were observed to enhance the effect of IFN-γ on the degree of T-cell apoptosis in HONE1 and AGS cells (Supplementary Fig. 17).

The Jurkat cell line is derived from an acute T-cell leukemia cell and is a commonly used investigation tool in T-cell research[29,30]. PMA and PHA were used to activate Jurkat cells and were then used to co-culture them with HONE1 cells at a ratio of 1:10. Similar results were obtained (Fig. 7f and Supplementary Figs. 18–21) when Jurkat cells were co-cultured with NPC cells. EBV-miR-BART11 and EBV-miR-BART17-3p induced apoptosis in Jurkat cells and inhibited cytokine production through FOXP1 and PBRM1 to upregulate PD-L1 expression.

**EBV-miR-BART11 and EBV-miR-BART17-3p promote tumor immune escape in mice with tumor cell xenografts.** To further verify the effects of EBV-miR-BART11 and EBV-miR-BART17-3p on tumor immune escape, $5 \times 10^6$ HONE1 and AGS cells were transfected with EBV-miR-BART11 and EBV-miR-BART17-3p mimics or negative control, and HONE1-EBV and AGS-EBV cells were transfected with EBV-miR-BART11 and EBV-miR-BART17-3p inhibitors or negative control. Transfected cells were injected into mice subcutaneously. After an observation period of 7 days, the formation of palpable tumors was noted and $5 \times 10^7$ T cells presented with tumor antigens derived from cell lysates were injected into the tail vein for subsequent experiments. To identify whether EBV-mediated tumor growth was PD-L1-dependent, the PD-L1 inhibitor (Atezolizumab) was injected into mice after injecting T cells (Supplementary Fig. 22a). The in vivo imaging results of small animals showed that T cells gradually gathered at the tumor-forming site; however, the fluorescence intensity of T cells in the BART11+BART17-3p mimics +T cells group was weaker than that in the NC+ T cells group (Fig. 8). The signal intensity of the BART11+BART17-3p inhibitors +T cells group was stronger than that of the NC+ T cells group. The difference in fluorescence intensity diminished after injection of the PD-L1 inhibitor (Atezolizumab, Fig. 8).

T cells were extracted from peripheral blood cells of nude mice 7 days after the tail vein injection of T cells. EBV-miR-BART11 and EBV-miR-BART17-3p mimics significantly increased the degree of apoptosis of CD8+ T cells as observed using flow cytometry (Fig. 9a and Supplementary Fig. 22b) and inhibited IFN-γ secretion, as evidenced using qRT-PCR and ELISA methods (Fig. 9b, c). The PD-L1 inhibitor (Atezolizumab) could weaken these functions of the miRNAs mimics and decrease the difference in IFN-γ secretion levels between the NC+ T cells group and the BART11+BART17-3p mimics+ T cells group. EBV-miR-BART11 and EBV-miR-BART17-3p inhibitors showed the opposite results in mice.

To identify whether EBV-miR-BART11 and EBV-miR-BART17-3p affected tumor formation in mice, we determined

the tumor volumes and body weights of the mice and found that the values of both these parameters were lesser in the T cell-treated group than those observed in the untreated group. There was no significant difference in tumor size and tumor weight after BART11 + BART17-3p activation or inhibition; however, the tumors in the NC + T cells group were significantly smaller than those in the BART11 + BART17-3p mimics +T cells group and were bigger than those in the BART11 + BART17-3p inhibitors +T cells group (Fig. 10a and Supplementary Fig. 22c–e). When the PD-L1 inhibitor (Atezolizumab) was injected into mice after tumor formation, the difference in tumor size and tumor weight diminished between the NC + T cells group and the BART11 + BART17-3p mimics+T cells group. It indicates that T cells attack tumor cells and cause tumor cell apoptosis but EBV-miR-BART11 and EBV-miR-BART17-3p weaken the killing effect of T cells; additionally, these EBV-encoded miRNAs mediate tumor growth that is PD-L1-dependent.

IHC and ISH experiments conducted using mice subcutaneous tumor tissue sections showed that FOXP1 and PBRM1 expression was significantly lower and that for PD-L1 was significantly higher in the BART11+BART17-3p and BART11+BART17-3p+ T cells group than those in the NC and NC+ T cells groups. The expression of CD8 in T cells and of the expression of cleaved Caspase-3 and cleaved-PARP in tumor cells were lower in the BART11+ BART17-3p+ T cells group than in the NC+ T cells group (Supplementary Fig. 22f, g). Moreover, EBV-miR-BART17-3p inhibitors had opposite results about the expression of these molecules of mice. This indicates that EBV-miR-BART11 and EBV-miR-BART17-3p can promote CD8+ T cell apoptosis in mice receiving adoptive T-cell therapy, resulting in reduced numbers of tumor cells killed by T cells. There were no differences between the NC+ T cells group and the BART11 + BART17-3p+ T cells group after subjection to the PD-L1 inhibitor (Atezolizumab) injection. This result suggests that EBV-miR-BART11 and EBV-miR-BART17-3p may inhibit the attack of T cells on tumor cells through the induction of a tumor immune escape mechanism via PD-L1 in mice.

## Discussion

T cells can recognize and eliminate cancerous cells and perform immune surveillance functions after the TCR binds to MHC molecules on antigen-presenting cells[31]. However, tumor cells often evade immune surveillance through various mechanisms[32–34]; for example, they show high PD-L1 expression and establish interactions with PD-1 on the surface of T cells to prevent the effective activation of tumor antigen-specific T cells, leading to tumor immune escape[8–10]. PD-L1 expression is regulated by multiple levels of pre-transcription, transcription, translation, and post-translational modification processes[8,35,36]. Among them, the transcriptional regulators, such as HIF1α, STAT1/STAT3, NF-κB, IRF1/IRF3, c-Myc, BRD4 can bind to the *PD-L1* promoter to upregulate PD-L1 expression[10]. Therefore, an in-depth study of the PD-1/PD-L1

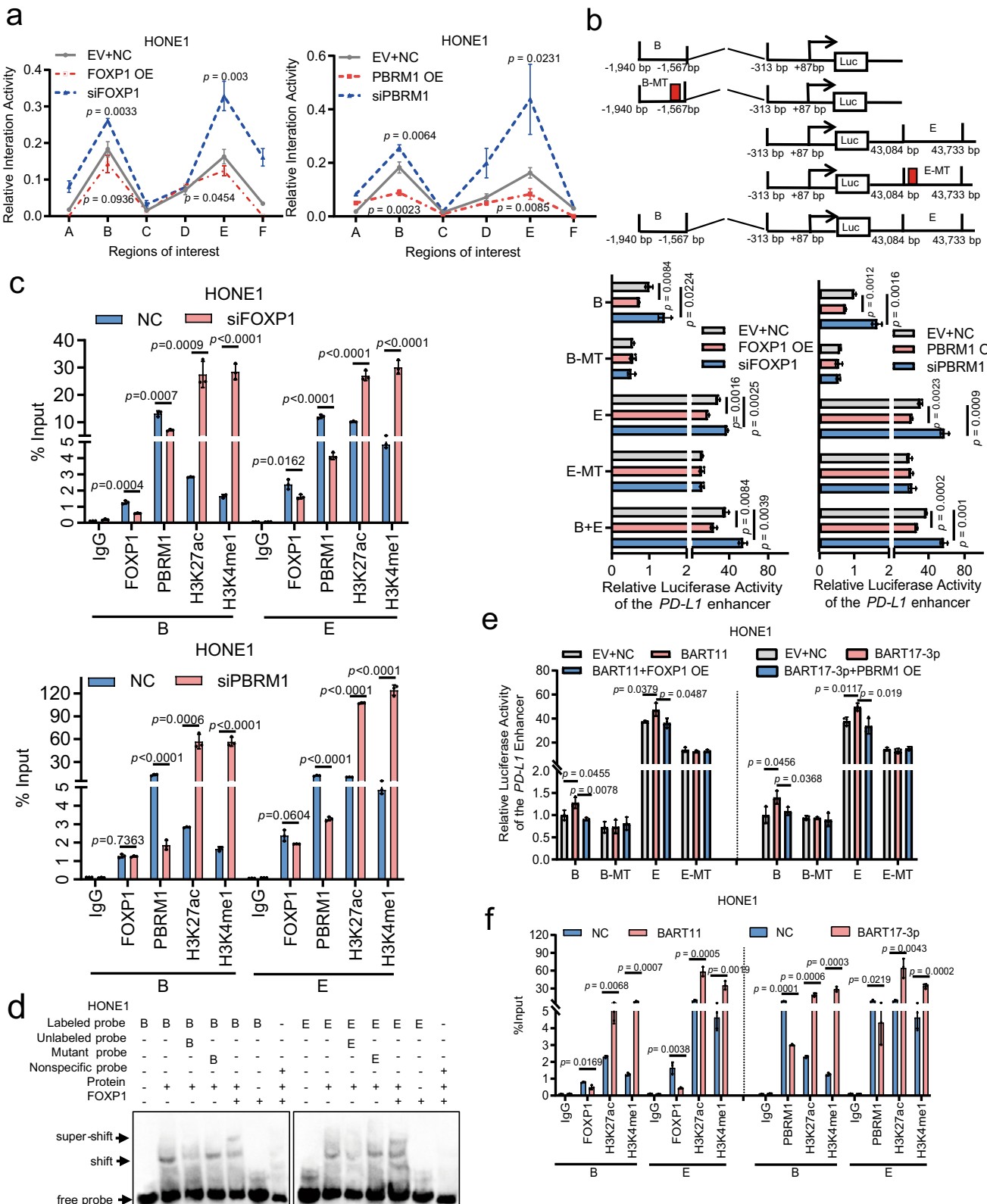

expression regulation mechanism is of considerable significance to improve its effectiveness in immunotherapy. This study demonstrate that the EBV-encoded miRNAs BART11 and BART17-3p target two transcriptional repressors, FOXP1 and PBRM1, that bind to the enhancer of *PD-L1*, thereby affecting the transcription and expression of PD-L1.

A variety of EBV protein products and EBV encoded circular RNAs regulate PD-L1 expression. EBV encoded LMP1 gene upregulates PD-L1 expression through regulating the JAK/STAT and AP-1-associated transcriptional activities[3]. LMP1 also promotes PD-L1 expression through the NF-κB pathway in NK/T cell lymphoma[4]. EBNA1 activates JAK2/STAT1/IRF-1 signals

**Fig. 5 FOXP1 and PBRM1 inhibit the transcription of PD-L1 via binding to the enhancer. a** The interaction frequency between the enhancer (A-F) and the promoter region of *PD-L1* was identified in HONE1 cells transfected with the overexpression vectors or siRNAs for *FOXP1* or *PBRM1*. The relative interaction frequency was normalized to the closest BamH I digestion site. **b** A series of wild type and mutant luciferase reporter vectors were prepared for the enhancer B and E regions of *PD-L1*. The reporter activity was measured in HONE1 cells co-transfected with the corresponding reporter vector and the FOXP1, or PBRM1 overexpression vector and siRNAs. **c** ChIP using the antibodies against FOXP1, PBRM1, H3K27ac, and H3K4me1 were performed to identify whether knockdown of *FOXP1* or *PBRM1* affected the binding of FOXP1 and PBRM1, and the H3K27ac and H3K4me1 modification in the *PD-L1* enhancers B and E in HONE1 cells. **d** EMSA assay was performed to detect whether FOXP1 could bind to the *PD-L1* enhancers B and E. Lane 1: only biotin-labeled probes were added; lane 2: nuclear protein and biotin-labeled probes were added; lane 3: nuclear protein, and biotin-labeled probes and competitively bound unlabeled probes were added in a ratio of 1:2; lane 4: nuclear protein, and biotin-labeled probes and mutant unlabeled probes in a ratio of 1:2 were added; lane 5: nuclear protein, biotin-labeled probes and anti-FOXP1 antibody were added simultaneously; lane 6: biotin-labeled probes and anti-FOXP1 antibody were added; lane 7: nuclear protein, nonspecific probe, and anti-FOXP1 antibody were added simultaneously. **e** The luciferase reporter activity of the *PD-L1* enhancers B and E via FOXP1 or PBRM1 was measured in HONE1 co-transfected with the corresponding reporter vector, EBV-miR-BART11 or EBV-miR-BART17-3p mimics, and the overexpression vector for FOXP1 or PBRM1. **f** ChIP was performed to examine whether EBV-miR-BART11 or EBV-miR-BART17-3p affected the binding of FOXP1 or PBRM1 and the H3K27ac and H3K4me1 modification in the *PD-L1* enhancers B and E after transfection of EBV-miR-BART11 or EBV-miR-BART17-3p mimics in HONE1. Data are presented as mean ± s.d, and *p*-values are calculated by unpaired two-sided *t*-test in **a–c**, **e**, **f**. Source data are provided as a Source Data file.

and induces *PD-L1* promoter activity[5]. EBNA2 inhibits miR-34a by downregulating EBF1 expression, thereby promoting PD-L1 expression[6]. EBV-miR-BHRF1-2-5p reduces PD-L1 expression through binding to *PD-L1* 3′-UTR in EBV-positive diffuse large B-cell lymphoma[37]. CircBART2.2 induced PD-L1 expression by binding with RIG-I protein and activating the RIG-I pathway, resulting in the immune escape of NPC cells[38]. EBV also plays a pivotal role in host immune escape through the encoded BART cluster miRNAs to subvert and evade host immune responses. EBV-miR-BART22 can target *LMP2A* and eventually escape the host immune response, promoting the survival of NPC cells[39]. EBV-miR-BART6-3p can bind to the *RIG-I* 3′-UTR, resulting in the inhibition of RIG-I-like receptor signaling and the type I IFN response[40]. These findings suggested that BART cluster miRNAs might play an important regulatory function in EBV-mediated tumor immune escape. However, there are no reports available on the study of BART cluster miRNAs regulating PD-L1 expression and the promotion of tumor immune escape. In this study, we investigated a mechanism for the regulation of PD-L1 expression through the PD-L1 enhancer. The mechanism is different from other viral molecules, and the potential mechanism underlying the cooperation between these viral miRNAs and other viral factors remains obscure.

The tumor microenvironment includes not only tumor cells but also cytokines and antigen-presenting cells (APCs) such as dendritic cells, and macrophages. These APCs inhibit T-cell activation by expressing PD-L1 and by establishing interactions with PD-1 on T cells, and concurrently, they can secrete a substantial number of cytokines such as IFN-γ, IL-2 and IL-10, which indirectly promote PD-L1 expression in tumor cells and APCs[41–43]. We found that EBV-encoded EBV-miR-BART11 and EBV-miR-BART17-3p can further enhance the effect of IFN-γ in promoting PD-L1 expression[5]. Whether EBV can regulate PD-L1 expression in APCs, which merits future study.

FOXP1 is a member of the FOXP subfamily, which is expressed in various tissues of the body and demonstrates a tumor suppressor effect in a variety of solid tumors. Previous studies have shown that FOXP1, as a transcription repressor, can inhibit the activity of target gene enhancers or promoters. It binds to the same forkhead binding sites in the *IL-7Rα* enhancer region and *IL-9* promoter region through the competitive binding of FOXO1, thereby inhibiting IL-7Rα and IL-9 expression[44]. Furthermore, FOXP1 is a transcriptional inhibitor of NF-κB signaling[45]; and NF-κB induces PD-L1 expression[10]. In this study, we found that FOXP1 could bind to the enhancer of *PD-L1* through two specific binding sites and could inhibit the transcriptional activity of the *PD-L1* gene. Additionally, FOXP1 can demonstrate a synergistic

function with FOXP3 to maintain the stability of Treg and immunosuppressive functions[46,47], and the inhibition of FOXP1 can activate T-cell killing functions and promote IFN-γ secretion[48]. Thus, these considerations will lay the foundation of future research and the analysis of the potential applicability of FOXP1 as a target for EBV-associated tumor immunotherapy.

PBRM1, also known as BAF180, is a subunit of the PBAF subtype of the SWI/SNF chromatin-remodeling complex. It affects histone modifications, such as H3K4me1, and regulates the transcription of the *IL-10* gene[17]. PBRM1 and the polycomb protein EZH2 directly bind to the *cis*-regulatory elements of *RIG-1* and *MDA5*, thereby inhibiting the transcription of target genes[18]. Moreover, knockout of *PBRM1* increases PD-L1 expression in tumor cells, although the specific mechanism has not been elucidated[49]. PBRM1-deficient cells show significantly increased expression of IFN-γ-responsive genes[50]; however, IFN-γ is almost not expressed in NPC cells. Our study has reported that PBRM1 can regulate PD-L1 expression in the absence of IFN-γ stimulation, indicating that PBRM1 can regulate PD-L1 expression without involvement of the IFN-γ pathway. As the intricate targeting of chromatin remodeling complexes remains poorly understood, we could not determine the DNA-binding sites of PBRM1; however, we found that FOXP1 could directly bind to the enhancer of PD-L1. Additionally, we also observed that PBRM1 established interactions with FOXP1 and co-localized with the PD-L1 enhancer.

Based on specific subunits, chromatin remodeling complexes can be divided into the BAF and PBAF subtypes. There are a few overlaps in certain components between the PBAF and BAF chromatin remodeling complexes[51]. PBRM1 functions as a component of PBAF and DPF2 is a specific subunit of BAF[28]. DPF2 is enriched in active enhancers and promotes transcription[52,53], while PBAF plays a role in transcriptional inhibition[28]. Competition exists between the two components because both can be enriched in the enhancer to regulate transcription[51]. Our results showed that FOXP1 could bind to the PBAF subtype to which PBRM1 belongs; however, it could not bind to the BAF subtype through interactions with PBRM1 and such a phenomenon was not observed with DPF2. PBRM1 and DPF2 compete with each other through subunit assembly and binding to the PD-L1 enhancer, thereby regulating PD-L1 expression. Our results showed the potential relationship between the BAF/PBAF complex and PD-L1 and provided insights into a research direction for future drug development involving the BAF/PBAF complex.

Enhancers are defined as DNA regulatory elements with strong transcriptional activation characteristics and play an important

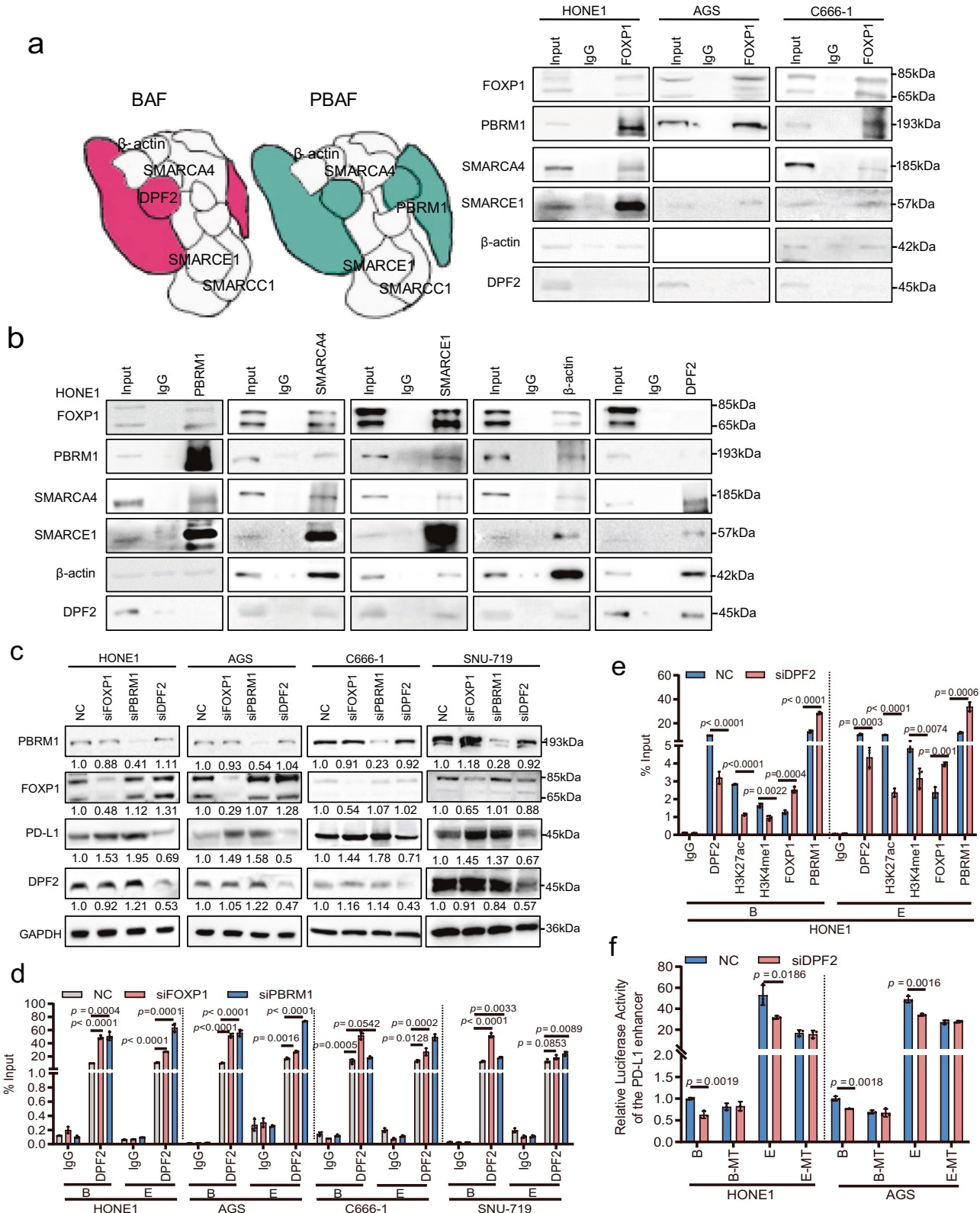

role in tumorigenesis, development, the immune response, and other processes[53–56]. A previous study has shown the presence of enhancers in the *PD-L1* transcriptional regulatory region[57]. Broad-spectrum inhibitors of super-enhancers, such as JQ1, can enhance the cytotoxicity of T cells by reducing PD-L1 expression[57,58]. Therapy involving the targeting of tumor-specific enhancers may be more effective in eliminating tumor cells while maintaining normal cell growth. Transcriptional regulatory factors are bound to the *PD-L1* promoter[10]. Our results showed that the transcription repressors FOXP1 and PBRM1 and the transcription activator DPF2 exerted effects on the *PD-L1* enhancers. Furthermore, the analysis of the regulation of the *PD-*

**Fig. 6 FOXP1 establishes interaction with the PBRM1 containing the PBAF complex and inhibits PD-L1 expression. a** Co-IP experiments were performed using an antibody against FOXP1 in HONE1, AGS, and C666-1 cells. FOXP1 establishes interactions with PBRM1, SMARCA4, SMARCE1, and β-actin in the PBAF complex, but does not establish interactions with DPF2 in the BAF complex. **b** Co-IP experiments were performed in HONE1 cells using PBRM1, SMARCA4, SMARCE1, β-actin, and DPF2 antibodies in HONE1 cells. PBRM1, SMARCA4, SMARCE1, and β-actin in the PBAF complex bind to FOXP1; DPF2 in the BAF complex does not bind to FOXP1. **c** Western blotting was performed to detect the expression of PD-L1, FOXP1, PBRM1, and DPF2 in HONE1, AGS, C666-1, and SNU-719 cells after knockdown of *FOXP1*, *PBRM1*, or *DPF2*. GAPDH was used as an internal control. **d** ChIP experiments using anti-DPF2 antibody were used to identify DPF2 binding to the *PD-L1* enhancers B and E in HONE1, AGS, C666-1, and SNU-719 cells after knockdown of *FOXP1* or *PBRM1*. **e** ChIP experiments using antibodies against DPF2, H3K27ac, H3K4me1, FOXP1, or PBRM1 were performed to identify whether *DPF2* knockdown affected the binding of DPF2, H3K27ac, H3K4me1, FOXP1, and PBRM1 in the *PD-L1* enhancers B, and E, and the H3K27ac, H3K4me1 modifications in HONE1 cells. **f** Luciferase reporter gene assays were performed to identify whether *DPF2* knockdown affected the luciferase activity of the *PD-L1* enhancers B and E in HONE1 and AGS cells. Data are presented as mean ± s.d., and *p*-values are calculated by unpaired two-sided *t*-test in **e**, **f**. Source data are provided as a Source Data file.

*L1* enhancers will help us devise better strategies to overcome tumor immune escape without affecting the physiological functions of normal cells.

Collectively, our study shows that EBV-miR-BART11 and EBV-miR-BART17-3p promote PD-L1 expression by targeting FOXP1 and *PBRM1*, respectively. FOXP1 establishes interactions with the PBAF subtype to which PBRM1 belongs, binds to the *PD-L1* enhancer, and promotes *PD-L1* transcription, resulting in tumor immune escape (Fig. 10b). These results provide insights into the transcriptional regulatory mechanism for *PD-L1*, and aid better understanding of the PD-L1 expression regulation in EBV-infected tumors and may help in achieving better design of therapeutic strategies with EBV-encoded miRNAs as combined or independent target. Targeting of EBV miRNAs (EBV-miR-BART11 and EBV-miR-BART17-3p) in combination with PD-L1 immunotherapy may improve the clinical outcome of EBV-infected cancer patients.

## Methods

**Clinical tissue samples**. For qRT-PCR detection, tissue samples obtained from 82 patients with NPC (40 EBV-negative; 42 EBV-positive) and non-tumor nasopharyngeal epithelial tissue samples collected from 31 individuals (Supplementary Data 2) were collected at the Cancer Hospital of Central South University from 2016 to 2019. For in situ hybridization (ISH) or immunohistochemistry (IHC), samples obtained from 52 patients with NPC (13 EBV-negative; 39 EBV-positive) and non-tumor nasopharyngeal epithelium tissue samples from 36 individuals (Supplementary Table 1) and samples obtained from 40 gastric adenocarcinoma tissues (15 EBV-negative, 25 EBV-positive) and 20 normal gastric mucosa tissues (Supplementary Table 2) were collected at the Second Xiangya Hospital of Central South University from 2016 to 2019. These tissue samples were confirmed by histopathological examination, and before use, informed consent of the patient was obtained as authorized by the Ethics Committee of Central South University.

**ISH and IHC**. The probes for ISH (TSINGKE, China) were synthesized and labeled with DIG at both 5' and 3' ends (Supplementary Table 3). Briefly, paraffin-embedded sections were deparaffinized and were treated with pepsin diluted in 3% citric acid for 15 min and 4% paraformaldehyde for 10 min. Then the sections were prehybridization and hybridized with 2 nM DIG-labeled EBER1 probe at 60 °C overnight[59] or 20 nM DIG-labeled EBV-miR-BART11-3p, EBV-miR-BART11-5p, EBV-miR-BART17-3p probe at 55 °C overnight.

IHC was performed using the streptavidin-peroxidase-complex method. The tissue slices were de-waxed and rehydrated, cooked with 1 mM EDTA containing antigen retrieval solution at 95 °C, cooled to room temperature, and subsequently blocked using an endogenous peroxidase blocker. Tissue sections were incubated with the primary antibody at 4 °C overnight. After washing with PBS, the sections were incubated with biotin-labeled secondary antibodies for 1 h at room temperature, followed by treatment with Streptomyces anti-biotin peroxidase solution (MXB, China) for 15 min. The antibodies are listed in Supplementary Table 4.

To evaluate the number of ISH- or IHC-positive cells, a semi-quantitative scoring criterion was used to estimate the staining intensity and positive areas. The staining intensity was scored from 0 to 3, based on the standards, indicating 0 (no staining), 1 (weak staining intensity), 2 (moderate or strong staining intensity with background colors), and 3 (strong staining intensity). The scoring was graded as 0 (negative), 1 (<25% positive), 2 (25%–50% positive), 3 (50%-75%), and 4 (>75% positive) as per the proportion of stained cells. The final score deduced from the multiplication of these two scores ranged between 0 and 12. All sections were

independently scored by two pathologists who were blinded to the clinicopathological features.

**Cell culture and transfection**. Cell lines including human EBV-negative immortalized normal nasopharyngeal epithelial cell line NP69, EBV-positive NPC cell line C666-1, EBV-negative NPC cell lines HONE1, HNE2, and CNE2, EBV-negative GC cell line AGS, and HONE1-EBV and AGS-EBV cell lines were constructed by stably transfecting EBV (Akata-derived BAC)[60] in EBV-negative HONE1 and AGS, EBV-positive human Burkitt lymphoma Akata cells, EBV-transformed marmoset leukocyte B95-8 cells, EBV-positive GC cell line SNU-719, and human T lymphocyte leukemia cell line Jurkat were maintained in RPMI 1640 medium (Gibco, USA) supplemented with 10% FBS (Gibco) at 37 °C and 5% CO$_2$. Normal gastric epithelial cell line GES-1 was maintained in DMEM medium (Gibco, USA) supplemented with 10% FBS (Gibco). To activate Jurkat cells, 10 nM phorbol-12-myristate-13-acetate (PMA; Sigma) and 50 uM phytohemagglutinin (PHA; Sigma) were used. HONE1-EBV cell line was a generous gift from Professor George Sai Wah Tsao, University of Hong Kong and Professor Xin Li, Southern Medical University. AGS-EBV cell line was donated by Professor Lunquan Sun from Xiangya Hospital of Central South University.

The pIRESneo3-FOXP1 (CDS, NM-032682) vector was previously constructed and stored in our laboratory; the pcDNA3.1-PBRM1 (CDS, NM-001350074) and the control vector were purchased from YouBioCo.

The non-target scrambled siRNA controls were provided by RIBOBIO Co. Two siRNAs for *FOXP1*, *PBRM1*, and *DPF2*, respectively, were synthesized (Supplementary Table 3).

For transfection, the HiPerFect transfection reagent (Qiagen, Germany) or the Neofect DNA transfection reagent (Neofect biotech, China) was used to transfect the EBV miRNA mimics, inhibitors, gene plasmids, or siRNAs as per the manufacturer's instructions.

**Quantitative real-time PCR (qRT-PCR)**. Total RNA was extracted from cells by Trizol reagent (Invitrogen) and cDNA samples were synthesized with the HiScript II 1st Strand cDNA Synthesis Kit (Vazyme). Stem-loop real-time qRT-PCR for mature miRNAs was done with the Qiagen QuantiTect SYBR Green PCR Kits (Qiagen) and qRT-PCR for mRNA expression was performed using the ChamQ SYBR Color qPCR Master Mix (Vazyme), according to the manufacturer's instructions. Real-time PCR was run on CFX Real-Time PCR Detection System with CFX Manager TM software, version 3.1 (Bio-Rad). The electrophoresis of the qRT-PCR products were taken by the chemiluminescence imaging system with Quantity One software (Bio-Rad). The relative quantification method ($2^{-\Delta\Delta Ct}$) was used to calculate the fold change in gene expression using three biological replicates for the qRT-PCR. The primers are listed in Supplementary Table 3.

**Western blotting**. Cells were lysed with RIPA buffer (Beyotime, China) containing a proteinase inhibitor cocktail (Keygen, China) and phosphatase (Beyotime). Protein sample concentrations were estimated using the BCA assay kit (Bio-Rad, USA). Protein samples were separated using a 10% SDS-PAGE gel and the separated proteins were subsequently transferred to a PVDF membrane (Millipore, USA). The membrane was blocked using 5% milk for 1 h at room temperature and was probed with primary antibodies at 4 °C overnight. Next, the membranes were incubated with the peroxidase-conjugated secondary antibody for 1 h at room temperature. GAPDH was used as the protein loading control. The blots were developed using the eECL Western Blot Kit (CWBIO Technology, China) and imaged using a chemiluminescence imaging system MiniChemi™ 610 with Sage-Capture™ software (SAGECREATION, China). The antibodies used are listed in Supplementary Table 4.

**EBV preparation and infection**. Resuscitate Akata or B95-8 cells, gradually increase the culture medium to the required number of milliliters, the density is $2–3 × 10^6$ cells per milliliter, and add goat anti-human IgG at 0.75% (v/v) concentration at 37 °C for 6 h. After starving the cells for 4–7 days, the cell supernatant

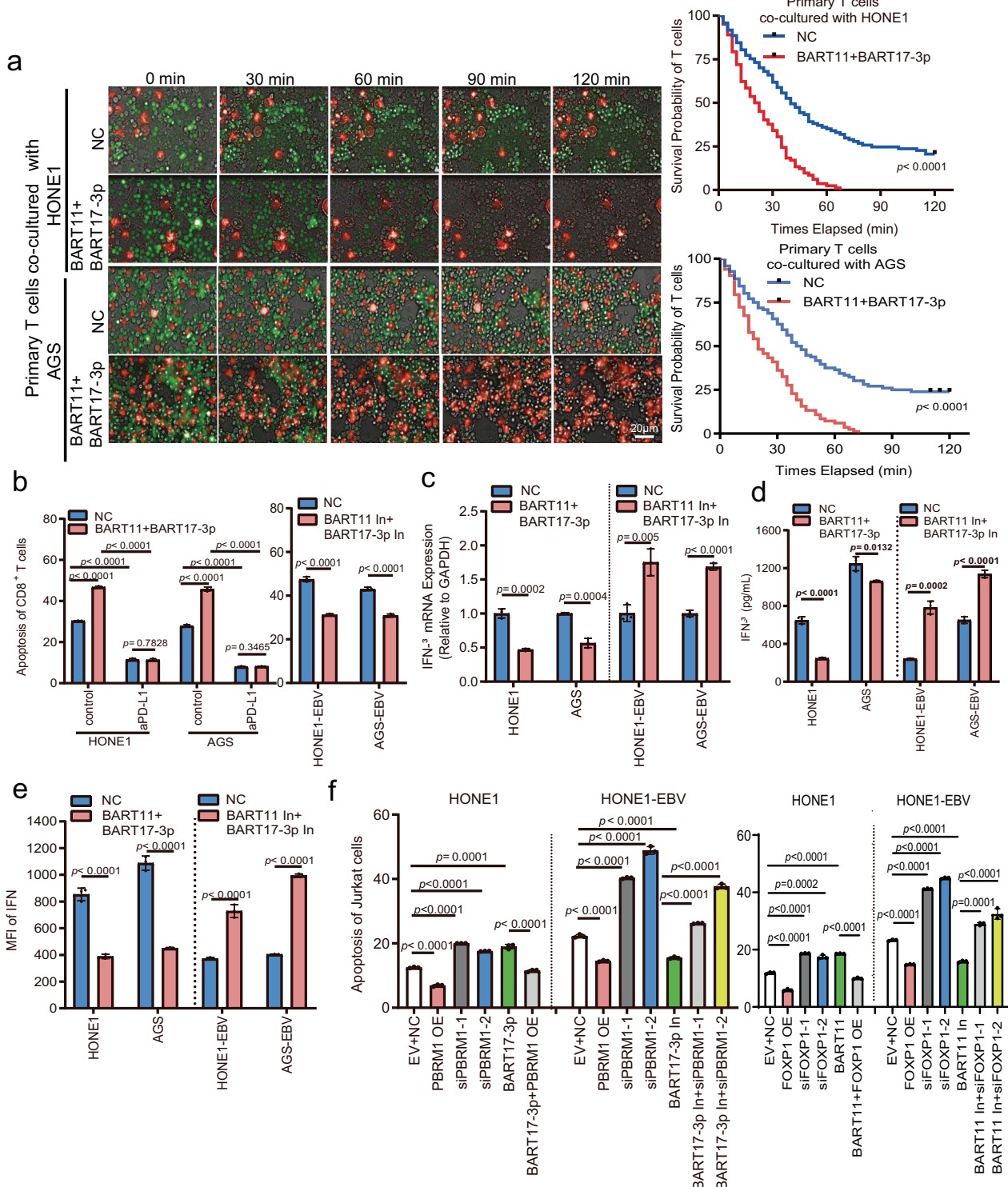

was collected. Freeze and thaw three times, centrifuge at $1500 \times g$ for 15 min. The supernatant was filtered through a 0.45 μm filter membrane to remove cell debris. The virus was concentrated by centrifugation at $60,000 \times g$ for 2 h in an ultra-centrifuge, and stored at $-80\ °C$ after aliquoting. Five-hundred microliters purified EBV virions were added to each well to infect EBV-negative cells at $37\ °C$, $5\%\ CO_2$ for 3 h in a 12-well plate. Add enough medium to continue culturing for another 1 day[60–62].

**Luciferase reporter assay**. For PBRM1, the cells were co-transfected with EBV-miR-BART17-3p mimics or inhibitors, The luciferase reporter vector (*PBRM1*-WT or *PBRM1*-MT), and the pRL-TK Renilla luciferase vector (Promega, USA). For the

PD-L1 enhancer and promoter analysis, cells were co-transfected with luciferase reporter vectors constructed with the *PD-L1* promoters or enhancers, pRL-TK Renilla luciferase vector, and EBV-miR-BART11, EBV-miR-BART17-3p mimics or inhibitors, the FOXP1 or PBRM1 expression vectors, or siRNA. Luciferase activity was measured using the Dual-Luciferase Reporter Assay System (Promega) by plate reader with SoftMax® Pro 7 software, version 7.1.0 (MolecularDevices).

**RNA immunoprecipitation (RIP)**. The cells were subjected to washing steps twice with PBS and were lysed using the RIPA lysis buffer (Thermo Fisher) containing a protease inhibitor cocktail (Keygen). The lysate was subsequently incubated with magnetic beads conjugated with anti-AGO2 or the rabbit IgG antibody control.

**Fig. 7 EBV-miR-BART11 and EBV-miR-BART17-3p induce T cell apoptosis by promoting PD-L1 expression. a** A high-content screening system was used to track the activity status of primary T cells co-cultured with HONE1 or AGS cells after overexpression of EBV-miR-BART11 and EBV-miR-BART17-3p; living tumor cells: red (CM-Dil); living T cells: green (CMFDA); living and apoptotic cells: bright field. Image (left): superimposed signals for red and green fluorescence and bright field; image (right): the statistical graph according to the T-cell signaling intensity (CMFDA). Magnification: ×400; scale bars = 20 μm. **b** Flow cytometry to identify whether blockade of PD-L1 using anti-PD-L1 antibody affects EBV-miR-BART11 and EBV-miR-BART17-3p regulation in primary T-cell apoptosis. T cells were co-cultured with HONE1 or AGS cells after overexpression of EBV-miR-BART11 and EBV-miR-BART17-3p, or HONE1-EBV or AGS-EBV after inhibition of EBV-miR-BART11 and EBV-miR-BART17-3p. The sum of Annexin V-positive and PI-positive cells in each group was counted as the proportion of T cells undergoing apoptosis. $n = 3$ biologically independent samples. The original results are shown in Supplementary Figs. 13 and 14. **c** qRT-PCR performed to identify the *IFN-γ* mRNA level in primary T cells. T cells were co-cultured with HONE1 or AGS after overexpression of EBV-miR-BART11 and EBV-miR-BART17-3p or HONE1-EBV or AGS-EBV after inhibition of EBV-miR-BART11 and EBV-miR-BART17-3p. *GAPDH* was used as an internal control. $n = 3$ biologically independent samples. **d**, **e** ELISA and flow cytometry analysis of IFN-γ secretion from primary T cells co-cultured with HONE1 or AGS cells after overexpression of EBV-miR-BART11 and EBV-miR-BART17-3p, or HONE1-EBV, or AGS-EBV after the inhibition of EBV-miR-BART11 and EBV-miR-BART17-3p. $n = 3$ biologically independent samples. The MFI for IFN-γ in each group was calculated by flow cytometry analysis and the original results are shown in Supplementary Fig. 15. **f** Flow cytometry analysis of the apoptotic status of Jurkat T cells. The cells were co-cultured with HONE1 or HONE1-EBV cells after the overexpression or inhibition of FOXP1 or PBRM1 and EBV-miR-BART11 or EBV-miR-BART17-3p. $n = 3$ biologically independent samples. The original results are shown in Supplementary Figs. 19 and 21. Data are presented as mean ± s.d., in **b–f**. Moreover, *p*-values are calculated by unpaired two-sided *t*-test in **a–f**. Source data are provided as a Source Data file.

---

The magnetic beads were washed twice with high-salinity wash buffer (700 mM NaCl). The immunoprecipitated RNA was separated using the TRIzol reagent and analyzed via qRT-PCR.

**Immunofluorescence.** Cells were cultured at ~50% density in 24-well chamber slides and subsequently fixed with 4% paraformaldehyde for 10 min at room temperature. Next, the cells were permeabilized with 0.2% Triton X-100 and blocked with 5% BSA for 30 min at room temperature. The primary antibodies were diluted in the blocking buffer and incubated in the slides at 4 °C overnight. Secondary antibodies were added to the samples and incubated at room temperature for 30 min. After 4′, 6-diamidino-2-phenylindole (DAPI) staining, samples were treated with an anti-fade mounting medium and sealed using a coverslip. The cells were imaged using a confocal laser scanning microscope (UltraView Vox; Perkin-Elmer, USA) with Volocity software, version 6.1.1 (PerkinElmer).

**Immunoprecipitation.** Cells were cultured in 10 cm dishes for 48 h after transfection and were harvested using an IP lysis buffer (50 mM Tris [pH 7.4], 150 mM NaCl, 1% NP-40, 0.1% SDS) containing a proteinase inhibitor cocktail (Keygen). Protein lysates were incubated on a rotator with 1 μg of primary antibodies for 2 h at room temperature, followed by the addition of 30 μL of IP beads (Bimake, China) and incubation on a rotator at 4 °C overnight. The beads and immune complexes were subjected to washing steps with IP lysis buffer 5 times, with each wash involving rotation at 20 s per round for 5 min at room temperature. The samples were boiled in SDS loading buffer at 95 °C for 5 min. The immunoprecipitated samples were detected via western blotting.

**Chromatin immunoprecipitation (ChIP).** ChIP was performed with a ChIP Assay Kit (Millipore) according to the manufacturer's procedure. Briefly, $5 \times 10^6$ cells were transfected in a 10 cm culture dish for 48 h, cross-linked using 1% paraformaldehyde for 10 min at room temperature, and incubated with glycine for 5 min to stop the cross-linking reaction. The fixed cells are harvested, lysed, and sonicated. The lysates were incubated with 50 μL of protein A/G magnetic beads (Bimake) and 5 μg of primary antibody on a rotator at 4 °C overnight. The DNA-protein cross-links were eluted and proteins were digested with proteinase K (Beyotime). The digested DNA was purified and pelleted using the DNA Purify Kit (Millipore) and amplified by qRT-PCR.

**Chromatin conformation capture (3 C).** Nuclear proteins were prepared according to the ChIP protocol. 500 U of the BamHI restriction enzyme (TaKaRa, Japan) was added to the DNA-protein cross-linked sample and digested overnight at 37 °C. The next day, 10% SDS was added to the digested samples, which were incubated at 65 °C for 20 min to inactivate the restriction enzyme. The sample was diluted 1:10 using the DNA Ligation Kit (TaKaRa), incubated at 16 °C for 4.5 h, and further incubated at room temperature for 30 min. Next, 300 mg of proteinase K (Beyotime) and 300 mg of RNase A (Beyotime) was added to the sample and incubated at 65 °C overnight. Then digested DNA was purified and pelleted using the DNA Purify Kit (Millipore) and amplified by qRT-PCR.

**Electrophoretic mobility shift assay (EMSA).** The probes for EMSA were synthesized and were prepared labeled or unlabeled with biotin at the 5' end (Supplementary Table 3). The DNA binding reaction was performed using the Chemiluminescent EMSA Kit (Beyotime) as per the manufacturer's instructions. Unlabeled probes were added simultaneously as competitors with the labeled probes. To identify DNA-binding proteins, nuclear extracts from HONE1 cells

were incubated with 1 μg of antibody against FOXP1 or PBRM1 at room temperature for 20 min before the addition of biotin-labeled probes. The samples were electrophoresed on a 4% polyacrylamide gel, transferred to a nylon membrane (Millipore), and crosslinked using UV-light. After blocking, the nylon membranes were incubated with the Streptavidin-HRP conjugate for 15 min on a shaker at room temperature and was imaged using a chemiluminescence imaging system (SAGECREATION).

**RNA pull-down assays.** The RNA pull-down assay was performed using the Magnetic RNA Protein Pull-Down Kit (Thermo Fisher, USA) according to the manufacturer's instructions. the specific EBV miRNA probe labeled with biotin was transfected into cells. Tumor cell lysates were added to the beads along with protein-RNA binding buffer (0.2 M Tris [pH 7.5], 0.5 M NaCl, 20 mM MgCl$_2$), and incubated overnight at 4 °C on a rotator. The beads were washed twice, the immunoprecipitated RNA was separated using the TRIzol reagent (Invitrogen), and the samples were analyzed by qRT-PCR.

**Liquid-chromatography coupled to tandem mass spectrometry (LC-MS/MS).** Purified protein complex eluates were concentrated using IP beads (Bimake), resolved by SDS-PAGE, and stained with Coomassie Brilliant Blue (Beyotime). Stained bands that differed from the control were excised, subjected to in-gel reduction, alkylated, and digested with trypsin (Gibco) at 37 °C overnight. The digested peptides were dried and resuspended in an MS-compatible buffer, and the mixture was analyzed using the LTQ Orbitrap Velos MS (Thermo Fisher) in combination with the UltiMate RSLC Nano LC system (Dionex). The Proteome Discoverer 1.4 software (Thermo Fisher) was used to identify the protein, and the files were imported and used to explore the UniProtKB/Swiss-Prot database. The mass tolerances of precursors and fragments were set to 10 ppm and 0.8 Da, respectively. Data on the peptides with a false discovery rate of <1% ($q < 0.01$) were discarded.

**Generation of dendritic cells (DCs) and T lymphocytes.** To achieve antigen presentation in tumor cells and to generate specific cellular immunity against the cell line, we obtained DCs and T cells. Based on the manufacturer's instructions, peripheral blood mononuclear cells (PBMCs) were obtained from the peripheral blood of healthy donors using the Ficoll-Hypaque method (Cytiva, USA). For the generation of DCs, the isolated PBMCs were added to 10% FBS-1640 medium containing 50 ng/mL GM-CSF and 20 ng/mL IL-4 (Sino Biological, China), and were cultured for 5 days. Cell differentiation was monitored using light microscopy. To promote DC maturation, 25 ng/mL interferon γ (IFN-γ; Sino Biological) was added for incubation for 1 day and co-culture with HONE1 cell lysates was performed for a period of 1 day. T cells were subjected to expansion in vitro by adding CD3/CD28 MACSiBead (Miltenyi, Germany) and 15 ng/mL IL-2, 5 ng/mL IL-7, and 10 ng/mL IL-15 (Sino Biological) to PBMCs and by incubating for 8 days. To generate tumor-specific T cells, the prepared DCs and the expanded T cells were co-cultured at a 1:5 ratio for 5 days in the medium supplemented with IL-2, IL-7, and IL-15. Fresh medium and cytokines were replaced with the fresh medium every 2 days during the experimental period.

**Induction of apoptosis in T cells by tumor.** After transfection for 24 h, the treated cells were seeded on a glass bottom, light-resistant 24-well plate for 12 h, subjected to staining with the live-cell fluorescent dye CM-Dil (Thermo Fisher), and co-cultured with activated human primary T cells or Jurkat cells labeled with the CMFDA dye (Thermo Fisher) at a ratio of 1:10. A high-content screening system with Harmony software, version 4.9 (PerkinElmer, USA) or a confocal microscope

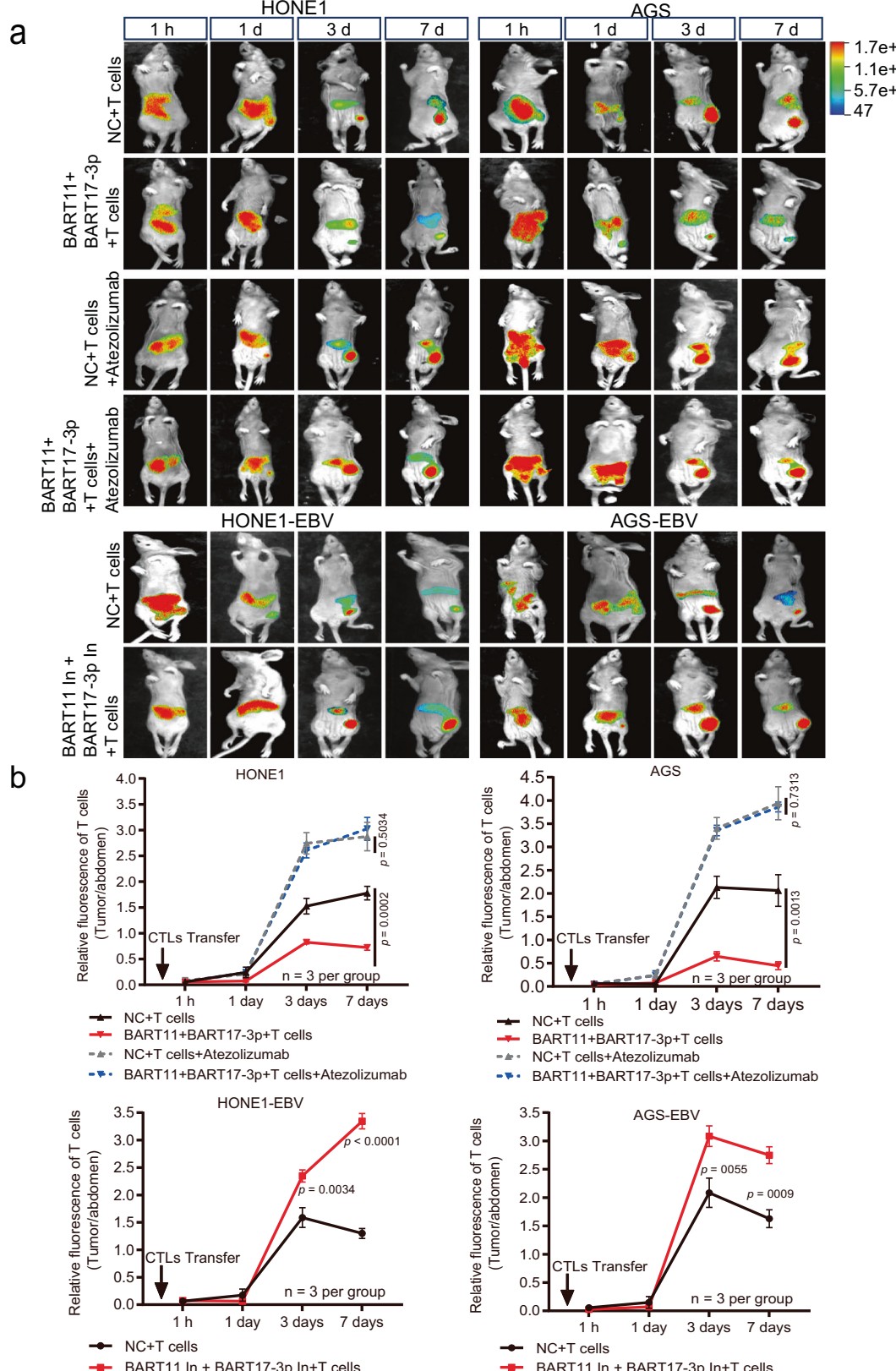

**Fig. 8 Accumulation of DiR-stained T cells in xenograft mice models after injection of T cells.** Nude mice were injected with tumor cells after transfection of EBV-miR-BART11 and EBV-miR-BART17-3p mimics, inhibitors, or negative control (NC). After following an observation period of 7 days, DiR-labeled activated T cells were injected through the tail vein. The PD-L1 inhibitor (Atezolizumab) was injected into mice after performing T cells injection. **a** Representative images of the DiR fluorescence in mice. The color scales indicate the DiR fluorescence intensity in mice. **b** Statistical results according to the DiR fluorescence signal ratio in mice collected at 1 hour, 1 day, 3 days and 7 days post-injection (the DiR fluorescence signal ratio, T-cell fluorescence intensity in tumors of the root of the right thigh: T-cell fluorescence intensity in the abdomen of mice). Data are presented as mean ± s.d., and $p$-values are calculated by unpaired two-sided $t$-test. Source data are provided as a Source Data file.

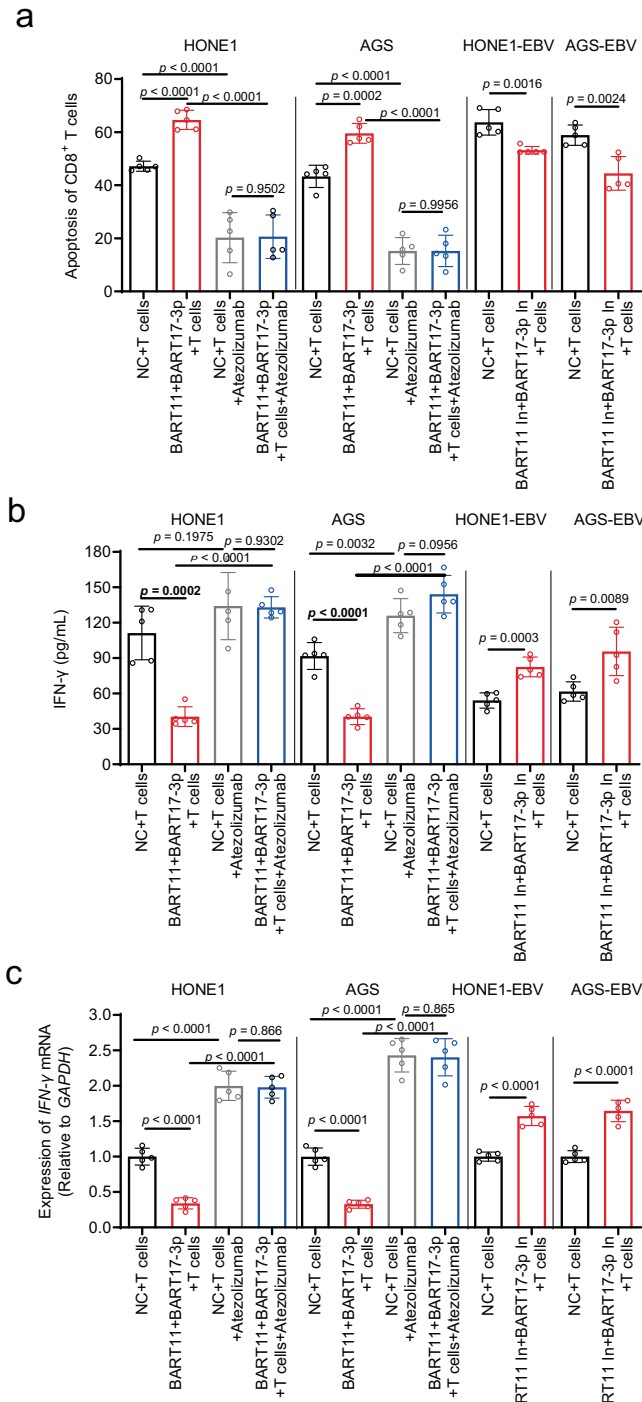

with Volocity software, version 6.1.1 (PerkinElmer) was used to track the T-cell fluorescence status in real-time. Green fluorescence was observed in surviving T cells, and all living and apoptotic cells could be observed in bright fields.

**Flow cytometry.** The transfected tumor cells and activated human primary T cells were co-cultured in a 24-well plate for 3 h at 1:10. According to the manufacturer's instructions, use the Apoptosis Detection Kit (BD Pharmingen) and fluorescently labeled CD3 and CD8 antibodies to detect T cell apoptosis by flow cytometry. This kit can stain Annexin V and PI. The percentage of apoptotic cells includes the percentage of early (Annexin V+, PI−) and late apoptotic cells (Annexin V+, PI+). As mentioned above, with the addition of the Golgi transport blocker 25 ng/mL BFA (Selleck), the tumor cells and T cells were co-cultured for 3 h. Then T cells were collected and stained with cell membrane antibodies CD3, CD8. After pre-treatment with a membrane breaker, the fluorescently labeled cytokine IFN-γ antibody was added to incubate, and the IFN-γ in the T cells was detected by flow cytometry. Flow cytometry was performed using the DxPAthenaTM flow cytometer (Cytek, USA) with FlowJo CE software (Treestar, USA). The FlowJo v10 software (Treestar) was used for data analysis.

**Enzyme-linked immunosorbent assay (ELISA).** The transfected tumor cells were co-cultured with activated T cells for 2 h or with Jurkat cells for 12 h. The culture supernatant was collected to detect cytokines; alternately, the serum of mice treated with T cells was obtained. The ELISA kit (Boster Bio) was used to detect IL-2, GZMB, and IFN-γ by plate reader with SoftMax® Pro 7 software, version 7.1.0 (MolecularDevices). All experiments were performed as per the manufacturer's instructions.

**Tumor cell-derived xenograft (CDX) implantation and adoptive T-cell transfer therapy in mice.** Mice (Balb/c Nude, 4 weeks old, female) were obtained from the Laboratory Animal Center of Central South University and maintained under SPF conditions in a controlled environment of 20–22 °C, with a 12/12 h light/dark cycle, 50–70% humidity, and food and water provided ad libitum. All procedures were approved by the Ethics Committee of Central South University. HONE1 or AGS cells ($5 \times 10^6$) were transfected with EBV-miR-BART11 and EBV-miR-BART17-3p mimics, and HONE1-EBV or AGS-EBV cells ($5 \times 10^6$) were transfected with EBV-miR-BART11 and EBV-miR-BART17-3p inhibitors[7,38]. Then, the transfected cells were injected into the right thigh root of each mouse. Tumor formation was observed macroscopically 7 days later. After palpable tumor formation, $5 \times 10^7$ T cells presenting with the tumor antigens derived from cell lysates were intravenously transfused into mice via the tail vein[29,63]. The PD-L1 inhibitor (5 mg/kg, Atezolizumab) was injected to block the PD-L1/PD-1 immune checkpoint and to reduce immunosuppressive signals found within the tumor microenvironment, which consequently increased T cell-mediated immunity against the tumor[64–66].

For the first set (three mice per group), the small animal in vivo imaging system (Bruker, USA) was used to evaluate human primary T cells distribution, which were labeled using the Deep Red live cell fluorescent dye (DiR, ThermoFisher). For the second set (five mice per group), mouse peripheral blood was extracted for performing qRT-PCR, ELISA, or flow cytometric analysis after 7 days of adoptive T-cell treatment without the fluorescent dye. For the third set (eight mice per group), the mice were euthanized when their tumor size exceeded 10% of their body weight, or they had lost 20% of their weight, or met the institutional euthanasia standards for their overall health condition;the remaining mice were sacrificed after 25 days of T-cell injection; the tumors were excised, weighed, and photographed, following which tumor tissues were embedded in paraffin and 4 µm sectioned. The weight and the tumor size of the remaining mice were measured regularly. The tumor volume was calculated using the formula $0.52 \times L \times W^2$, where $L$ and $W$ represent the long and short diameters of the tumor, respectively. A laboratory technician responsible for animal care and measurement of tumor growth was blinded to the group allocation during all animal experiments and outcome assessment. The sections were subjected to IHC and ISH analyses.

**Live-cell imaging of T cells.** The activated primary T cells were subjected to staining procedures with DiR (Thermo Fisher) and were injected into the CDX nude mice via the tail vein. After the mice were anesthetized using 3% isoflurane (RWD, China), the small animal in vivo imaging system (Bruker, USA) was used to dynamically observe the accumulation of T cells at the tumor-forming site.

**Statistics and reproducibility.** All statistical analyses were performed using the GraphPad Prism 8 software (GraphPad, USA). The Student's t-test was used to evaluate the significant difference between two groups of data. The relationship between PD-L1 expression and clinical characteristics was analyzed by using the F-test. The correlation analysis was performed by linear regression. All experiments were repeated at least biological triplicate. Data are expressed as mean ± standard deviation (SD). $p < 0.05$ was considered statistically significant.

**Fig. 9 EBV-miR-BART11 and EBV-miR-BART17-3p promote T-cell apoptosis and IFN-γ secretion in xenograft mice models. a** Flow cytometry analysis performed to detect T-cell apoptosis in the CDX nude mice model. Activated T cells were injected into nude mice, and after 7 days, peripheral blood samples were extracted for flow cytometry. $n = 5$ per group. The original results are shown in Supplementary Fig. 22b. **b** qRT-PCR analysis of *IFN-γ* mRNA levels in peripheral blood after injection of activated T cells. $n = 5$ per group. *GAPDH* was used as an internal control. **c** ELISA for the IFN-γ secretion in peripheral serum after injection of activated T cells. $n = 5$ per group. Data are presented as mean ± s.d, and $p$-values are calculated by unpaired two-sided t-test in **a**–**c**. Source data are provided as a Source Data file.

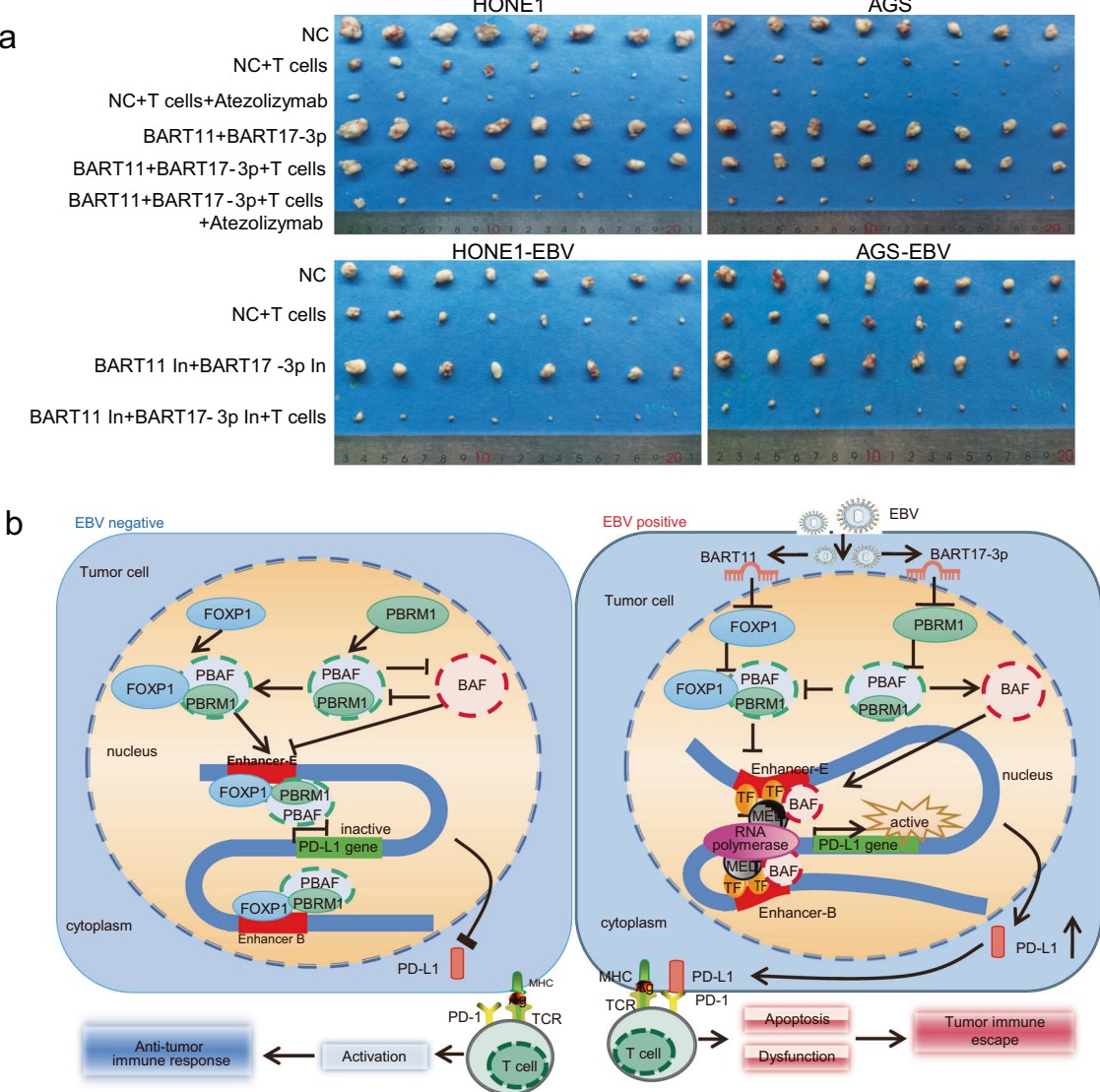

**Fig. 10 EBV-miR-BART11 and EBV-miR-BART17-3p promote tumor escape in xenograft mice models. a** Bright-field images of transplanted tumors in the CDX nude mice models. The mice were sacrificed after 25 days of injection with the activated T cells. $n = 8$ per group. **b** Mechanism of EBV-miR-BART11 and EBV-miR-BART17-3p in the activation of *PD-L1* and the promotion of tumor immune escape. EBV-encoded BART11 or BART17-3p target *FOXP1* or *PBRM1* to transcriptionally repress their expression, respectively. FOXP1 binds directly to the *PD-L1* enhancer, whereas PBRM1 cannot directly bind to it. PBRM1 forms the PBAF complex and binds to FOXP1 to exert effects on the *PD-L1* enhancer. The PBAF and BAF complexes compete with each other in assembly to bind with the *PD-L1* enhancer to affect PD-L1 expression, resulting in tumor immune escape. Source data are provided as a Source Data file.

**Reporting summary**. Further information on research design is available in the Nature Research Reporting Summary linked to this article.

## Data availability

Data referenced in this study are available in The Cancer Genome Atlas (https://tcga-data.nci.nih.gov/docs/publications/stad_2014/), Gene Expression Omnibus with the accession code GSE12452, GSE65801, GSE51575, GSE32960, GSE36682, GSE64634, and GSE95749, JASPAR CORE database, the 9th release (2022) (https://jaspar.genereg.net/), ENCODE database displayed in UCSC Browser (http://genome.ucsc.edu/cgi-bin/hgTracks?db=hg19&lastVirtModeType=default&lastVirtModeExtraState=&virtModeType=default&virtMode=0&nonVirtPosition=&position=chr9%3A5430000%2D5500000&hgsid=1226038355_xQJAKzHP0remahAa3CmCprv7ihQT). The mass spectrometry proteomics data have been deposited to the ProteomeXchange Consortium (http://proteomecentral.proteomexchange.org) via the PRIDE (https://www.ebi.ac.uk/pride/) partner repository with the dataset identifier PXD031014. Source data are provided with this paper.

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

## Acknowledgements

We thank Prof. George Sai Wah Tsao, Prof. Xin Li and Prof. Lunquan Sun for providing cell lines. This work was supported in part by grants from the National Natural Science Foundation of China (81772928 [W.X.], 82073135 [W.X.], U21A20382 [W.X.], 81972776

[Z.Y.Z.], and U20A20367 [Z.Y.Z.]), the Natural Science Foundation of Hunan Province (2019JJ50872 [L.S.], and 2020JJ4125 [F.Y.W.]).

## Author contributions

J.W. designed the project and completed most experiments. J.S.G., Y.A.W., J.Y.G. performed some of the experiments. F.X., X.J.J., X.Y.D., Z.J.G., S.S.Z., Q.J.Y., Y.H., X.Y.L., L.S. collected tissue samples. L.S.Z. analyzed the ChIP-Seq data. J.W. analyzed the data and wrote the manuscript. C.G., F.Y.W., Z.L., M.Z., B.X., Y.L., W.X., and Z.Y.Z. revised the manuscript. W.X. and Z.Y.Z. are responsible for research supervision and funding acquisition. All authors read and approved the final manuscript.

## Competing interests

The authors declare no competing interests.

## Ethics approval and consent to participate

The tissue samples informed consent of the patients and health donors were obtained as authorized by the Ethics Committee of Central South University before use.
