## [Peer Review File · Nature Communications]

EBV miRNAs BART11 and BART17-3p promote immune escape through the enhancer-mediated transcription of PD-L1REVIEWER COMMENTS

Reviewer #1 (Remarks to the Author):

Wang et al. investigated the mechanisms regulating PD-L1 expression and underlying tumor immune escape in EBV-associated NPC and GC. They infected EBV negative NPC cell lines HNE2 and CNE2 with Akata-derived and B95-8-derived EBV viruses; PD-L1 expression was significantly promoted in Akata-derived EBV-infected cells, but not in B95-8-derived EBV-infected cells. EBV B95-8 strain lacks the 12 kb BART locus where several BART miRNAs are encoded. The BART cluster miRNAs in the B95-8 deletion region were therefore investigated, and EBV-miR-BART17-3p, EBV-miR-BART11-3p, and EBV-miR-BART11-5p were found to regulate PD-L1 expression. EBV-miR-BART11 targeted FOXP1, and EBV-miR-BART17-3p targeted PBRM1. FOXP1 and PBRM1 were found to bind to the super-enhancer of PD-L1 to inhibit its expression. It was thus found that EBV-miR-BART11 and EBV-miR-BART17-3p enhanced the transcription of PD-L1 via downregulation of FOXP1 and PBRM1, resulting in the promotion of tumor immune escape, providing potential new targets for EBV-related tumor immunotherapy.

General comments:

While PD-L1 expression is known to be elevated in EBV-positive NPC and GC, the specific mechanisms of EBV-dependent promotion of PD-L1 expression has not been fully clarified. This paper gain insight to such mechanisms, and provide potentially interesting data. Although EBV-miR-BART11 and EBV-miR-BART17-3p seem to play a critical role in EBV-positive NPC and GC, several experiments were not conducted for both cancer types, causing uncertainty. Concerns to be addressed are raised as below.

Specific comments:

1. EBV-negative and EBV-positive NPC cell lines, and experimentally EBV-infected NPC cell line were analyzed for NPC in this study. But only AGS and EBV-infected AGS cells were analyzed for GC (throughout the manuscript). EBV-positive GC cell lines established from primary EBV-associated GC tissue should be additionally analyzed, as C666-1 was analyzed for NPC.
2. Expression levels were generally analyzed for NPC tissue samples, but not for GC tissues (throughout the manuscript). GC tissues should also be analyzed and shown.
3. siRNA experiments were generally done by using only one siRNA against a target gene (throughout the manuscript). At least two siRNAs must be analyzed for each gene, considering off-targets of siRNA.
4. In comparison of Akata-derived and B95-8-derived EBV viruses (Fig. S1), HNE2 and CNE2 cells were used as EBV negative NPC cell lines. But HONE1 was used for the subsequent experiments. The comparison of Akata-derived and B95-8-derived EBV viruses should also be conducted for HONE1.
5. In Fig. 6d-f, Enhancer-B and E were described as “SEs B and E”. The title also describes “super-enhancer”. But the authors did not show enough data to support that these enhancers are super-enhancers. The authors should add necessary ChIP-seq analysis, or simply change “super-enhancer” to “enhancer”.
6. The fluorescence intensity was compared and analyzed in Fig. 7a, but the quantitative levels should be analyzed and shown to demonstrate “significant” difference.
7. NPC xenograft was analyzed, but GC xenograft should also be analyzed.

Reviewer #2 (Remarks to the Author):

This is a technically impressive and expansive study of the role of EBV miRNAs in regulating PD-L1. There is extensive analysis of the target identification (PBRM1, FOXP1), mechanism of gene regulation by PBRM1 and FOXP1, and the effects of PD-L1 on T-cell control of tumorigenesis in mouse models. Overall, the technical quality of the experiments is extremely high, although a few concerns are raised in the specific comments. The extensive study leaves little doubt that the

mechanism occurs and has functional consequences for regulation of PD-L1 in EBV+ tumor cells, and T-cell response to these changes in PD-L1. Another general concern is that many viral factors target PD-L1 gene expression, and it is not clear how these viral miRNAs work together with other viral factors, namely EBNA2 and LMP1.

Specific Comments:

1. Fig1a. The difference in PD-L1 expression between EBV+ and EBV- NPC seems small despite a weak statistically significant difference.
2. Fig 4B. The colocalization of FOXP1 and PBRM1 needs additional information. Are these endogenous proteins or expressed ectopically? Since the proteins are both pan-nuclear, it is not correct to argue they colocalize, except that they are both pan-nuclear.
3. Fig. S8C. Are patients with NPC studied after radiation treatment? This may account for the immunosuppression of CD3 T-cells. Also, it is not clear how this relates to NPC tumor microenvironment effects of PD-L1.
4. Fig. S9A. Luciferase analysis of the PD-L1 regulatory regions is not convincing and difficult to interpret.
5. Fig 5C. ChIP assay requires additional negative controls showing that FOXP1 and PBRM1 do not bind to a different, non-specific DNA region. This is necessary to conclude that binding is specific for these DNA sites (B and E).
6. Fig. 5D, EMSA is not convincing. Additional controls are needed, such as addition of FOXP1 antibody alone and with non-specific probe.
7. Fig. 7B. An important control is to show that BARTs miRNAs have no effect on T-cell in the absence of any co-cultured cells. In other words, do BART miRNAs have any direct effects on the T-cells.
8. General concern: A number of EBV factors have been implicated in regulating PD-L1, including LMP1. How does the effects of LMP1 (<https://doi.org/10.1182/blood.V128.22.4134.4134>) and EBNA2 (PMID: 29946193) compare to viral BART miRNAs for modulating PD-L1 transcription. Is it the same mechanism, or complementary?

Reviewer #3 (Remarks to the Author):

Wang et al. observed that PD-L1 is highly expressed in Epstein-Barr virus (EBV)-positive nasopharyngeal carcinoma (NPC) and gastric cancer (GC) tissues. To further clarify the mechanisms regulating PD-L1 expression, Wang et al. infected the EBV-negative NPC cell lines with B95-8-derived EBV viruses, which lack several BART miRNAs, and found no change in PD-L1 expression. Hence, transfection of miRNA and its inhibitor further demonstrated that BART17-3p, BART11 (BART11-3p, BART11-5p) could regulate PD-L1 expression. Using plasmid, miRNA, and inhibitor of miRNA, the authors investigated that miR-BART17-3p and miR-BART11 upregulate PD-L1 expression by targeting PBRM1 and FOXP1, respectively. PBRM1 formed a complex with FOXP1 and PBAF, and this complex inhibits PD-L1 transcription by binding to its super-enhancer B and E regions. In vivo study showed that HONE1 cells expressing both miR-BART11 and miR-BART17-3p weakened T cells' killing effect accompanied with PD-L1 expression. The regulatory functions of EBV-encoded BART cluster miRNAs in PD-L1 had been elaborated thoroughly. However, it is regrettable that no in vivo study using inhibitors of miR-BART11 and miR-BART17-3p to decrease the expression of PD-L1 and tumor size.

Major comment:

1. The animal study in Figure 8 only indicates EBV miRNAs-induced tumorigenesis is T cell sensitive. It remains elusive if EBV-mediated tumor growth is PD-L1 dependent.
2. Lack of genetic evidence weakens the claim of PD-L1 in EBV miRNAs tumorigenesis. Can EBV induce NC tumor growth in PD-L1 knockout background?
3. Inducible PD-L1 seems to be more critical for T cell suppression. Unfortunately, it was not mentioned throughout the manuscript.
4. The author delicately showed a novel mechanistic study on PD-L1 via super-enhancer; however, there is no strong indication of the therapeutic nor diagnostic value in this study.
5. Immune-competent mice would be a better model to test the efficacy of anti-PD-L1 in EBV miRNAs-mediated tumor progression.

Minor comment:

1. In Figures 1c and 1d, the C666-1 cell line has over 150 fold PD-L1 mRNA expression but only 4 fold expression shown in western blot. The problem is also found in Figures 1e and 1f.
2. Many scale bars in the Figures are redundant. Label only one scale bar in one of the images if the magnification is the same. (ex: Fig. 1b, 4b,d; Fig. 7a; Fig. 8g, etc.). Some scale bars are hardly read (Fig. 1b). The authors may consider changing the color.
3. Why use 29 or 19 μm as a scale bar in figures (ex: Fig.4b; Fig. 7a; Fig. S4, etc.)?
4. Some sentences are not completely shown in PDF (ex: PD-L1 TRS7(-1940bp--1567bp)- in Table S2; the title of Table S3).
5. Table S1a to S1d should be listed as Table S1 to S4.

Point-by-point responds to the reviewer's comments

We are grateful for the referees' valuable comments on our manuscript. We have carefully considered all the comments and questions raised by the reviewers. We took time to plan and carry out additional experiments, which helped to address the reviewers' comments and questions. Newly obtained data are included in the revised manuscript or supplementary information and the relevant discussions have been amended in the manuscript. We sincerely thank the reviewers for raising relevant questions and constructive comments which have, in our opinion, greatly helped us to improve the quality of the present work. We have outlined our responses to the points below.

Reviewer #1

1. EBV-positive GC cell lines established from primary EBV-associated GC tissue should be additionally analyzed, as C666-1 was analyzed for NPC.

Response: Thank you for your suggestion. We have added information on a series of experiments conducted using SNU-719, which is an EBV-positive GC cell line established from primary EBV-associated GC tissue. Normal gastric epithelial cell line GES-1 was used as the control.

2. GC tissues should also be analyzed and shown.

Response: Thank you for your suggestion. GC tissue expression levels have been analyzed and depicted in Fig. 1c, Fig. 4d, and Fig. S8c-S8d.

3. At least two siRNAs must be analyzed for each gene, considering off-targets of siRNA.

Response: Thank you for your suggestion. We have added information on a series of experiments to the study, including an analysis involving two siRNAs. Two siRNAs were designed and used for each target (FOXP1, PBRM1, and DPF2). Of them, one siRNA of each was selected in the 3'-UTR to rule out the effect of off-targets.

4. The comparison of Akata-derived and B95-8-derived EBV viruses should also be conducted for HONE1.

Response: Thank you for your suggestion. Akata-derived and B95-8-derived EBV viruses were compared and tests were conducted using HONE1 (Fig. S1d).

5. In Fig. 6d-f, Enhancer-B and E were described as “SEs B and E”. The title also describes “super-enhancer”. But the authors did not show enough data to support that these enhancers are super-enhancers. The authors should add necessary ChIP-seq analysis, or simply change “super-enhancer” to “enhancer”.

Response: Thank you for your suggestion. To identify whether the PD-L1 enhancer is the super-enhancer or not, the online ChIP-Seq data (GSE95749 and GSE89128) were downloaded and the Rank Ordering of Super-Enhancers (ROSE) algorithm (https://bitbucket.org/young_computation/rose) [1-2] was used to define active super-enhancers from H3K27Ac peaks in NPC cell line HNE1 and gastric cancer cell line N87. The union of H3K27Ac peaks within 12.5 kb distance was stitched together. The stitched domains were ranked by H3K27Ac signal, and super-enhancers were separated from typical enhancers by determining the point along the X-axis at which a line with a slope of 1 was tangent to the curve. Enhancers were then assigned to the transcript whose TSS was nearest the center of the enhancer. The results showed that the tangent to the peak points of the H3K27Ac signal in the PD-L1 enhancer regions B and E was less than 1, so these regions were the enhancers of PD-L1 rather than the super-enhancers (Figures below). We have revised “SEs B and E” and have changed “super-enhancer” to “enhancer”.

Identification of PD-L1-specific enhancers in NPC HNE1 cells and GC N87 cells. Total 1170 SEs in HNE1 and 729 SEs in N87 were identified above a flex-point (slope >1) according to the H3K27Ac signal.

- [1] Lovén, J., et al. Selective inhibition of tumor oncogenes by disruption of super-enhancers. *Cell* **153**, 320–334 (2013).
- [2] Whyte, WA., et al. (2013). Master transcription factors and mediator establish super-enhancers at key cell identity genes. *Cell* **153**, 307–319 (2013).

6. The fluorescence intensity was compared and analyzed in Fig. 7a, but the quantitative levels should be analyzed and shown to demonstrate “significant” difference.

Response: Thank you for your suggestion. The fluorescence intensity was calculated and illustrated using a graph provided in the right panel of Fig. 7a according to data on the live-cell signaling of T cells (CMFDA). We have also analyzed the fluorescence intensity depicted in the images of Fig. 7a (shown below).

7. NPC xenograft was analyzed, but GC xenograft should also be analyzed.

Response: Thank you for your suggestion. The GC xenograft has been analyzed as illustrated in Fig. 8-10, Fig. S22b-S22f.

Reviewer #2

The extensive study leaves little doubt that the mechanism occurs and has functional consequences for regulation of PD-L1 in EBV+ tumor cells, and T-cell response to these changes in PD-L1. Another general concern is that many viral factors target PD-L1 gene expression, and it is not clear how these viral miRNAs work together with other viral factors, namely EBNA2 and LMP1.

Response: Thank you for your suggestion. Many viral factors target PD-L1 gene expression. EBV miRNAs may perform in synergy with other viral factors, namely EBNA2 and LMP1, to regulate PD-L1 expression and to promote tumor immune escape. However, the potential mechanism underlying this cooperation remains obscure. We have discussed this aspect in the Discussion section, stating that this aspect merits further study.

Specific Comments:

1. Fig1a. The difference in PD-L1 expression between EBV+ and EBV- NPC seems small despite a weak statistically significant difference.

Response: Thank you for your suggestion. We have added information on new NPC

samples and have repeated this experiment, as shown in Fig. 1a. The data showed that there was a difference in PD-L1 expression between EBV⁺ and EBV⁻ NPCs.

2. Fig 4B. The colocalization of FOXP1 and PBRM1 needs additional information. Are these endogenous proteins or expressed ectopically? Since the proteins are both pan-nuclear, it is not correct to argue they colocalize, except that they are both pan-nuclear.

Response: Thank you for your suggestion. We have revised the figures (Fig. 4b) to depict endogenous localization confirmed using an anti-FOXP1 mouse monoclonal antibody (sc-398811, Santa Cruz) and an anti-BAF180/ PBRM1 rabbit polyclonal antibody (382286, ZENBIO).

3. Fig. S8C. Are patients with NPC studied after radiation treatment? This may account for the immunosuppression of CD3 T-cells. Also, it is not clear how this relates to NPC tumor microenvironment effects of PD-L1.

Response: Thank you for your suggestion. Though radiotherapy can cause T cell immunosuppression in patients with NPC, all NPC tissues in our study were collected from primary patients with NPC who had not undergone radiotherapy or chemotherapy before sample collection.

4. Fig. S9A. Luciferase analysis of the PD-L1 regulatory regions is not convincing and difficult to interpret.

Response: Thank you for your suggestion. We have revised information on the luciferase analysis of the PD-L1 regulatory regions, as depicted in Fig. S9a, to provide more convincing data. The data showed that the region from -1,940 to -1,567 bp was critical for the PD-L1 promoter to be subjected to regulation by PBRM1 and FOXP1.

5. Fig 5C. ChIP assay requires additional negative controls showing that FOXP1 and PBRM1 do not bind to a different, non-specific DNA region. This is necessary to conclude that binding is specific for these DNA sites (B and E).

Response: Thank you for your suggestion. Information on additional negative

controls (A, C, and D) for the ChIP assay has been added (Fig. S10d), and results showed that FOXP1 and PBRM1 did not bind to different, non-specific DNA regions (A, C, and D).

6. Fig. 5D, EMSA is not convincing. Additional controls are needed, such as addition of FOXP1 antibody alone and with non-specific probe.

Response: Thank you for your suggestion. We have incorporated data on additional controls, including the FOXP1 antibody alone and the antibody with a non-specific probe for EMSA (Fig. 5D and Fig. S10e).

7. Fig. 7B. An important control is to show that BARTs miRNAs have no effect on T-cell in the absence of any co-cultured cells. In other words, do BART miRNAs have any direct effects on the T-cells.

Response: Thank you for your suggestion. We have included data on flow cytometry experiments (Fig. S16) to show that BART miRNAs exert no effect on T cells in the absence of any co-cultured cells.

8. General concern: A number of EBV factors have been implicated in regulating PD-L1, including LMP1. How does the effects of LMP1 (<https://doi.org/10.1182/blood.V128.22.4134.4134>) and EBNA2 (PMID: 29946193) compare to viral BART miRNAs for modulating PD-L1 transcription. Is it the same mechanism, or complementary?

Response: Thank you for your suggestion. Many viral factors target PD-L1 gene expression. EBV miRNAs may function in synergy with other viral factors, namely EBNA2 and LMP1, to regulate PD-L1 expression and to promote tumor immune escape. LMP1 activates PD-L1 transcription mainly through the MAPK/NF- κ B Pathway [1]. EBNA2 promotes the expression of PD-L1 via early B-cell factor 1 (EBF1), a repressor of miR-34a transcription [2]. In this study, we investigated a new mechanism regulating PD-L1 expression through the PD-L1 enhancer. The potential mechanism underlying this cooperation between these viral miRNAs and other viral factors remains unclear. These viral factors may function in synergy or may establish

crosstalk with each other to regulate PD-L1 expression in EBV-infected tumors. We have added this description to the Discussion section, stating that this aspect merits further study.

[1] Bi, X., et al. PD-L1 is upregulated by EBV-driven LMP1 through NF- κ B pathway and correlates with poor prognosis in natural killer/T-cell lymphoma. *Blood* **128**,4134 (2016).

[2] Anastasiadou, E., et al. EBV-encoded EBNA2 alters immune checkpoint PD-L1 expression by downregulating miR-34a in B-cell lymphomas. *Leukemia* **33**,132-47 (2019).

Reviewer #3

It is regrettable that no *in vivo* study using inhibitors of miR-BART11 and miR-BART17-3p to decrease the expression of PD-L1 and tumor size.

Response: Thank you for your suggestion. We have added an *in vivo* study that was conducted using inhibitors of miR-BART11 and miR-BART17-3p to decrease the expression of PD-L1 . The data showed that use of both miR-BART11 and miR-BART17-3p inhibitors enhanced the T cells' killing effect and decrease the tumor size (Fig. 8-10).

Major comment:

1. The animal study in Figure 8 only indicates EBV miRNAs-induced tumorigenesis is T cell sensitive. It remains elusive if EBV-mediated tumor growth is PD-L1 dependent.

Response: Thank you for your suggestion. We have added information on an *in vivo* study (Fig. 8-10) conducted using a PD-L1 inhibitor (Atezolizumab) to show that EBV-mediated tumor growth is PD-L1-dependent. The PD-L1 inhibitor (Atezolizumab, Ambole, Cat. No. M6101) is a potent PD-1/PD-L1 interaction inhibitor; it binds to human PD-L1 and blocks its interaction with PD-1. Furthermore, it reduces immunosuppressive signals found within the tumor microenvironment and,

consequently, increases T cell-mediated immunity against the tumor. In the present study, it was observed that the use of Atezolizumab could weaken the effects of the miRNA mimics on tumor immune escape in the xenograft mice models.

2. Can EBV induce NPC tumor growth in PD-L1 knockout background?

Response: Thank you for your question. In human cells, we have identified that *FOXPI* and *PBRM1* are targets of miR-BART11 and miR-BART17-3p, respectively. However, the three EBV miRNAs cannot be used to target mice *Foxp1* and *Pbrm1* genes, and no mice NPC cell line is available. Verification of EBV-induced NPC tumor growth in a PD-L1 knockout background is challenging. Therefore, we used the PD-L1 inhibitor (Atezolizumab) to enforce a blockade on its interaction with PD-1 in the animal model and to determine whether EBV could induce tumor growth under PD-L1 inhibition (Fig. 8-10) conditions. Our data showed that use of the PD-L1 inhibitor (Atezolizumab) could weaken the effects of the miRNA mimics on tumor immune escape in the xenograft mice models.

3. Inducible PD-L1 seems to be more critical for T cell suppression. Unfortunately, it was not mentioned throughout the manuscript.

Response: Thank you for your suggestion. We have added descriptions of a few experiments that are conducted to identify T cell suppression by inducible PD-L1 after IFN- γ treatment using flow cytometric analysis (Fig. S17). The data showed that IFN- γ induced PD-L1 expression and enhanced the effect of EBV-miR-BART11 and EBV-miR-BART17-3p exerted on the degree of T-cell apoptosis in HONE1 and AGS cells.

4. The author delicately showed a novel mechanistic study on PD-L1 via super-enhancer; however, there is no strong indication of the therapeutic nor diagnostic value in this study.

Response: Thank you for your suggestion. We have added statements relevant to the therapeutic and diagnostic value of PD-L1 to the Discussion section.

5. Immune-competent mice would be a better model to test the efficacy of anti-PD-L1 in EBV miRNAs-mediated tumor progression.

Response: Thank you for your suggestion. In human cells, we have identified that the human *FOXP1* and *PBRM1* genes are the targets of miR-BART11 and miR-BART17-3p, respectively. However, the three EBV miRNAs cannot be used to target the mouse *Foxp1* and *Pbrm1* genes, and no mice NPC cell line is available. Verification of EBV-induced NPC tumor growth in a PD-L1 knockout background is challenging. Therefore, we have used the PD-L1 inhibitor (Atezolizumab) to enforce a blockade on its interaction with PD-1 in the animal model and to determine whether EBV can induce tumor growth under PD-L1 inhibition (Fig. 8-Fig. 10) conditions.

Minor comment:

1. In Figures 1c and 1d, the C666-1 cell line has over 150 fold PD-L1 mRNA expression but only 4 fold expression shown in western blot. The problem is also found in Figures 1e and 1f.

Response: Thank you for your suggestion. We have repeated these experiments and revised them as Fig. 1d and Fig. 1e.

2. Many scale bars in the Figures are redundant. Label only one scale bar in one of the images if the magnification is the same. (ex: Fig. 1b, 4b,d; Fig. 7a; Fig. 8g, etc.). Some scale bars are hardly read (Fig. 1b). The authors may consider changing the color.

Response: Thank you for your suggestion. We have performed the necessary revisions and only the scale bars have been labeled in each figure.

3. Why use 29 or 19 μm as a scale bar in figures (ex: Fig.4b; Fig. 7a; Fig. S4, etc.)?

Response: Thank you for your suggestion. We have revised the figures.

4. Some sentences are not completely shown in PDF (ex: PD-L1 TRS7(-1940bp--1567bp)- in Table S2; the title of Table S3).

Response: Thank you for your suggestion. We have revised this to ensure that the complete information is shown.

5. Table S1a to S1d should be listed as Table S1 to S4.

Response: Thank you for your suggestion. We have revised the tables.

REVIEWER COMMENTS

Reviewer #1 (Remarks to the Author):

Wang et al. revised their manuscript. Comments raised by this reviewer (Reviewer 1) have been generally addressed, including addition of EBV-positive GC cell line SNU-719, GC tissue experiments, two siRNA experiments, and changes from super-enhancers to enhancers. Other additional experiments requested by other reviewers have also been performed, which is welcome addition. In the revised manuscript, however, there is critical confusion as below.

Protein expression level is confusing.

In Fig. S4, PBMR1 is highly expressed in EBV-negative HONE1 and AGS cells, while BART17-3p downregulates PBMR1 expression. PBMR1 is repressed in EBV-positive HONE1-EBV, AGS-EBV, C666-1, and SNU-719 cells, while BART17-3p inhibitor upregulates PBMR1 expression. These are clear demonstration.

In Fig. 2, however, PBMR1 levels are sometimes quite high in EBV-positive cells, and further downregulation by siRNA is demonstrated. But in some columns, low expression of PBMR1 and its increased expression by BART17-3p inhibitor are also demonstrated.

Histone status is confusing.

In Fig. S10h, % Input of H3K27ac and H3K4me1 in EBV-positive C666-1 and SNU-719 cells is generally high (approx. 20-60 %), and is dramatically decreased after BART inhibitor treatment. These are clear demonstration.

In Fig. S10d, however, % Input of H3K27ac and H3K4me1 in EBV-positive C666-1 and SNU-719 cells is quite low (approx. 2-3% !), which is similar to EBV-negative AGS cell, and is significantly increased after siPBMR1 treatment, similarly in both EBV-positive C666-1 and SNU-719 cells and EBV-negative AGS cell.

In Fig. S9b, public ChIP-seq data of HNE1 and N87 were shown. Since enhancer regions of PD-L1 should be active in EBV-positive cells, ChIP-seq data of EBV-positive cells must be demonstrated here. However, N87 is reportedly EBV-negative GC cell, and it is unclear whether HNE1 is EBV-negative or -positive NPC cell line. As a result, it is unclear whether Fig. S9b shows active or inactive state of enhancers B and E in these cells.

By these data, the reviewer is confused and it is unclear whether or how much level BART expression/EBV infection contributes to downregulation of PBMR1, activation of enhancer regions of PD-L1, and upregulation of PD-L1. These regulations, however, must be critical as the authors draw schematic figure in Fig. 10b, and the authors should somehow address these concerns. For example, authors might perform western blot using the same or similar conditions for all the samples. The authors might add description clearly on the effect of further downregulation of PBMR1 by siRNA in addition to EBV. Concerns on unstable and confusing % Input data of ChIP-PCR must be solved; the authors might carefully perform ChIP experiment again, or could perform ChIP-seq for H3K27ac and H3K4me1 of each cell condition, at least for untreated EBV-negative and EBV-positive cell lines, so that epigenetic status at the enhancer regions can be compared fairly.

Reviewer #2 (Remarks to the Author):

The authors have addressed all of my previous concerns.

Reviewer #3 (Remarks to the Author):

1. The authors added PD-L1 inhibitor in HONE1 and AGS animal model, but the experiment is complicated and difficult to follow.

2. The authors added statements relevant to the therapeutic and diagnostic value of PD-L1 to the Discussion section. However, they did not highlight them in the revised version, resulting in spend time searching. They should list (ex: line 537-549) or highlight it.

3. In the previous version, the C666-1 cell line has over 150 fold PD-L1 mRNA expression but only 4 fold expression shown in western blot. They just deleted the mRNA result without explaining it.

4. In new Figure 8, the blue color indication is missing in the HONE1 graph.

5. The author has a similar study published in July 2021 in Cancer Research entitled “Epstein-Barr virus-encoded circular RNA circBART2.2 promotes immune escape of nasopharyngeal carcinoma by regulating PD-L1.” If EBV induces PD-L1 expression through two miRNA and one circular RNA, then which one is more important? The authors need to coordinate their work in the discussion.

Point-by-point responds to the reviewers' comments

We thank the reviewers for their insightful and constructive comments, which have helped us to significantly improve this manuscript. We have addressed the comments made by the reviewers and have incorporated their suggestions into the revised manuscript, as detailed in the point-by-point response below.

Reviewer #1 (Remarks to the Author):

1. Protein expression level is confusing.

In Fig. S4, PBM1 is highly expressed in EBV-negative HONE1 and AGS cells, while BART17-3p downregulates PBM1 expression. PBM1 is repressed in EBV-positive HONE1-EBV, AGS-EBV, C666-1, and SNU-719 cells, while BART17-3p inhibitor upregulates PBM1 expression. These are clear demonstration.

In Fig. 2, however, PBM1 levels are sometimes quite high in EBV-positive cells, and further downregulation by siRNA is demonstrated. But in some columns, low expression of PBM1 and its increased expression by BART17-3p inhibitor are also demonstrated.

Response: Thank you for your comments. In Fig. 2d, the expression of PBM1 in the siRNA group was too low to be exposed and was liable to overexposure in the PBM1 overexpression group. Additionally, there was an antibody titer problem, which caused some confusion in the western blot results. To resolve this issue, we loaded extracts from various groups on the same membrane for western blotting experiments using EBV-positive cell lines. The results showed that PBM1 overexpression and knockdown reduced and increased PD-L1 expression, respectively. Further, the EBV-miR-BART17-3p inhibitor increased PBM1 expression and down-regulated PD-L1 expression, and PD-L1 expression was reversed while inhibiting PBM1 simultaneously (Figures below). We have replaced these figures in the manuscript.

2. Histone status is confusing.

In Fig. S10h, % Input of H3K27ac and H3K4me1 in EBV-positive C666-1 and SNU-719 cells is generally high (approx. 20-60 %), and is dramatically decreased after BART inhibitor treatment. These are clear demonstration.

In Fig. S10d, however, % Input of H3K27ac and H3K4me1 in EBV-positive C666-1 and SNU-719 cells is quite low (approx. 2-3% !), which is similar to EBV-negative AGS cell, and is significantly increased after siPBRM1 treatment, similarly in both EBV-positive C666-1 and SNU-719 cells and EBV-negative AGS cell.

Response: We have repeated the ChIP-PCR experiment (Fig. S10d), and the input percentages of H3K27ac and H3K4me1 in EBV-positive C666-1 and SNU-719 cells were higher (~20% - 60%) than that in EBV-negative AGS cells (~2% - 15%).

3. In Fig. S9b, public ChIP-seq data of HNE1 and N87 were shown. Since enhancer regions of PD-L1 should be active in EBV-positive cells, ChIP-seq data of EBV-positive cells must be demonstrated here. However, N87 is reportedly EBV-negative GC cell, and it is unclear whether HNE1 is EBV-negative or -positive NPC cell line. As a result, it is unclear whether Fig. S9b shows active or inactive state of enhancers B and E in these cells.

Response: We have downloaded and reanalyzed the H3K27ac and H3K4me1 modification data from the published ChIP-Seq datasets (GSE95749 for EBV-positive C666-1 and EBV-negative HNE1) [1-3]. The modifications H3K27ac and H3K4me1 in EBV-positive C666-1 and EBV-negative HNE1 were compared. The data showed that activated enhancers tend to be enriched for H3K27ac and H3K4me1. The H3K27ac and H3K4me1 peaks in the B and E regions in EBV-positive C666-1 were higher than the peaks in EBV-negative HNE1 (Fig. S9b, Figures below), indicating that B and E are activated enhancers in C666-1.

- [1] Wang, H., et al. Neuropilin 1 is an entry factor that promotes EBV infection of nasopharyngeal epithelial cells. *Nat Commun* **6**,6240 (2015).
- [2] Xiang, T., et al. Vasculogenic mimicry formation in EBV-associated epithelial malignancies. *Nat Commun* **9**,5009 (2018).
- [3] Zhang, H., et al. Epstein-Barr virus activates F-box protein FBXO2 to limit viral infectivity by targeting glycoprotein B for degradation. *PLoS Pathog* **7**,e1007208 (2018).

4. For example, authors might perform western blot using the same or similar conditions for all the samples. The authors might add description clearly on the effect of further downregulation of PBMR1 by siRNA in addition to EBV. Concerns on unstable and confusing % Input data of ChIP-PCR must be solved; the authors might carefully

perform ChIP experiment again, or could perform ChIP-seq for H3K27ac and H3K4me1 of each cell condition, at least for untreated EBV-negative and EBV-positive cell lines, so that epigenetic status at the enhancer regions can be compared fairly.

Response: Thank you for your valuable advice. As per your comments, we have performed western blot experiments using the same conditions in Fig. 2d and ChIP-PCR in Fig. S10d. Further, we downloaded and analyzed published ChIP-Seq data (GSE95749 for EBV-positive C666-1 and EBV-negative HNE1). The modifications H3K27ac and H3K4me1 were compared between EBV-negative HNE1 and EBV-positive C666-1 and the results showed the presence of greater levels of H3K27ac and H3K4me1 modifications in the B and E regions in the EBV-positive C666-1 cell line (Fig. S9b).

Reviewer #3 (Remarks to the Author):

The authors added PD-L1 inhibitor in HONE1 and AGS animal model, but the experiment is complicated and difficult to follow.

Response: Our animal experiment was designed keeping in mind the following considerations:

1) Mouse *Foxp1* and *Pbrm1* are not the targets of these three EBV miRNAs and because no cell line for mice nasopharyngeal carcinoma is available, we can use human nasopharyngeal carcinoma cell lines alone for this experiment.

2) If human cancer cells are injected into mice with normal immunity, they will be eliminated due to the immune response. Human cancer cells cannot survive in mice with normal immune functions. Therefore, we chose nude mice having T cell deficiency to perform this experiment, and the model has been tested successfully in our [1] and other laboratories [2].

3) We injected human T cells into nude mice. These T cells must undergo an antigen presentation reaction *in vitro* so that T cells can recognize and kill human cancer cells, which can maximize the simulation of tumor immune surveillance in the human body. This adoptive T cell therapy in the xenograft model was performed by Huang et al. [3] and Chheda et al [4] and has been tested successfully in our laboratory [1].

In this study, we used the PD-L1 inhibitor (Atezolizumab) to enforce a blockade on interaction with PD-1 and to determine whether EBV miRNAs could induce tumor growth under PD-L1 inhibition conditions in the animal model. Atezolizumab injection was performed as described by Zheng et al [5], Su et al [6], and Huang et al [7], who reported the use of adoptive immunotherapy combined with PD-L1 inhibitors in mouse models. We have described the experimental protocol in the Methods section and have cited these references in the text.

[1] Ge, J.S., et al. Epstein-Barr virus-encoded circular RNA circBART2.2 promotes immune escape of nasopharyngeal carcinoma by regulating PD-L1. *Cancer Research* **81**, 5074-5088 (2021).

[2] Hsu, C.Y., et al. The Epstein-Barr Virus-Encoded MicroRNA MiR-BART9 Promotes Tumor Metastasis by Targeting E-Cadherin in Nasopharyngeal Carcinoma. *PLoS Pathog* **2**,e1003974 (2014).

[3] Huang, D., et al. NKILA lncRNA promotes tumor immune evasion by sensitizing T cells to activation-induced cell death. *Nat Immunol* **19**, 1112-1125 (2018).

[4] Chheda, Z, S., et al. Novel and shared neoantigen derived from histone 3 variant H3.3K27M mutation for glioma T cell therapy. *J Exp Med* **215**, 141-157 (2018).

[5]Zheng, Y., et al. Combining MPDL3280A with adoptive cell immunotherapy exerts better antitumor effects against cervical cancer. *Bioengineered* **8**, 367-373 (2017).

[6] Su, S., et al. Checkpoint Inhibition Overcomes ADCP-Induced Immunosuppression by Macrophages. *Cell* **175**, 442-457 (2018).

[7] Huang, A., et al. A human programmed death-ligand 1-expressing mouse tumor model for evaluating the therapeutic efficacy of anti-human PD-L1 antibodies. *Sci Rep* **7**, 42687 (2017).

2. The authors added statements relevant to the therapeutic and diagnostic value of PD-L1 to the Discussion section. However, they did not highlight them in the revised version, resulting in spend time searching. They should list (ex: line 537-549) or highlight it.

Response: Thank you for your suggestion. We will highlight the newly obtained data.

3. In the previous version, the C666-1 cell line has over 150 fold PD-L1 mRNA expression but only 4 fold expression shown in western blot. They just deleted the mRNA result without explaining it.

Response: We have repeated the experiments and included them in Fig. 1d. We used the immortalized normal nasopharyngeal epithelium cell line NP69 to compare with the EBV positive NPC cell line C666-1. Expression of PD-L1 in NP69 is extremely low and C666-1 shows a 16-fold expression change as shown in the western blotting results. Real-time PCR is more sensitive than western blotting. We have added data from gastric cancer tissues and another gastric cancer cell line, and considering the layout of the figure, we moved the mRNA results to supplementary figures (Fig. S1c and Fig. S1d).

4. In new Figure 8, the blue color indication is missing in the HONE1 graph.

Response: Thank you for highlighting the oversight. The error has been rectified.

5. The author has a similar study published in July 2021 in Cancer Research entitled “Epstein-Barr virus-encoded circular RNA circBART2.2 promotes immune escape of nasopharyngeal carcinoma by regulating PD-L1.” If EBV induces PD-L1 expression through two miRNA and one circular RNA, then which one is more important? The authors need to coordinate their work in the discussion.

Response: Thank you for your suggestion. In our paper recently published in Cancer Research, we found that circBART2.2, which is a circular RNA encoded by EBV, induced PD-L1 expression by binding with RIG-I protein and activating the transcription factors IRF3 and NF- κ B, leading to the immune escape of NPC cells. In this study, EBV-miR-BART11 and EBV-miR-BART17-3p targeted the two PD-L1 gene transcriptional repressors FOXP1 and PBRM1, respectively, and regulated PD-L1 enhancers to affect PD-L1 expression. We believe both of these are important. The

mechanism underlying the activity of these two viral miRNAs is different from that of circBART2.2, although the details underlying the cooperation between these viral miRNAs and other viral factors such as circBART2.2 remain obscure. Enhancers have a wide range of transcriptional activation domains and can bind transcription factors, which can greatly enhance transcription and significantly improve gene transcription efficiency. They may function in synergy or may establish crosstalk with each other to regulate PD-L1 in EBV-infected tumors. We have added this description to the Discussion section and have highlighted the change.

REVIEWER COMMENTS

Reviewer #1 (Remarks to the Author):

The concerns raised by this reviewer (Reviewer #1) for the revised manuscript NCOMMS-21-03874A, have been adequately addressed, and the re-revised manuscript NCOMMS-21-03874B now seems acceptable to this reviewer.

Reviewer #3 (Remarks to the Author):

The authors' responses are clear. However, there are still some questions in the revised manuscript.

Major comment:

1. According to the authors' hypothesis, EBV-miR-BART11 and EBV-miR-BART17-3p can increase the PD-L1. In Figure S22f, HONE1-EBV should have a higher PD-L1 expression in NC, but the IHC result had no PD-L1 signal in the NC group. Please explain it.

Minor comment:

1. Some formats need to be carefully checked again. For example, it has an extra basket in line 102 "(Fig. S1d, Fig. 1e)."
2. A dot is missing in line 46.
3. In line 118, the authors mentioned that "IFN- γ (10 ng/ μ L) stimulation", is it correct? The concentration of IFN- γ seems too high, so please check it.
4. In Fig. S11a, it should be a better way to point out the protein location on SDS-PAGE by an arrow.

Point-by-point responds to the reviewers' comments

We thank the reviewers for their insightful and constructive comments, which have helped us to significantly improve this manuscript. We have addressed the comments made by the reviewers and have incorporated their suggestions into the revised manuscript, as detailed in the point-by-point response below.

Major comment:

According to the authors' hypothesis, EBV-miR-BART11 and EBV-miR-BART17-3p can increase the PD-L1. In Figure S22f, HONE1-EBV should have a higher PD-L1 expression in NC, but the IHC result had no PD-L1 signal in the NC group. Please explain it.

Response: Thank you for your valuable advice. We double checked our Figure S22f data and compared the expression of PD-L1 in NC (HONE1), NC(HONE1-EBV), EBV-miR-BART11 In and BART17-3p In (HONE1-EBV), and EBV-miR-BART11 In, and BART17-3p In +T cells (HONE1-EBV). We found that HONE1-EBV has a little bit higher PD-L1 expression in NC, compared with the expression in HONE1-NC (EBV-negative), EBV-miR-BART11 In and BART17-3p In (HONE1-EBV), and EBV-miR-BART11 In, and BART17-3p In +T cells (HONE1-EBV). In order to get a higher resolution image, we took pictures for NC (HONE1-EBV) again and replaced it with a better image in the text.

Minor comment:

Some formats need to be carefully checked again. For example, it has an extra basket in line 102 “((Fig. S1d, Fig. 1e).”

Response: Thank you for your advice. We have revised it.

2. A dot is missing in line 46.

Response: Thank you for your advice. We have revised it.

3. In line 118, the authors mentioned that “IFN- γ (10 ng/ μ L) stimulation”, is it correct?

The concentration of IFN- γ seems too high, so please check it.

Response: Thank you for your advice. We double checked it and it should be “IFN- γ (10 ng/mL) stimulation” and have revised it.

4. In Fig. S11a, it should be a better way to point out the protein location on SDS-PAGE by an arrow.

Response: Thank you for your advice. We have added it on the protein location of SDS-PAGE.